



# A full year of aerosol size distribution data from the central Arctic under an extreme positive Arctic Oscillation: Insights from the MOSAiC expedition

Matthew Boyer[1], Diego Aliaga[1], Jakob Boyd Pernov[2], Hélène Angot[2], Lauriane L. J. Quéléver[1], Lubna Dada[2], Benjamin Heutte[2], Manuel Dall'Osto[3], David C. S. Beddows[4], Zoé Brasseur[1], Ivo Beck[2], Silvia Bucci[5], Marina Duetsch[5], Andreas Stohl[5], Tiia Laurila[1], Eija Asmi[6], Andreas Massling[7], Daniel Charles Thomas[7], Jakob Klenø Nøjgaard[7,11], Tak Chan[8], Sangeeta Sharma[8], Peter Tunved[9], Radovan Krejci[9], Hans Christen Hansson[9], Markku Kulmala[1], Tuukka Petäjä[1], Mikko Sipilä[1], Julia Schmale[2], Tuija Jokinen[1,10]

[1]Institute for Atmospheric and Earth System Research (INAR)/Physics, Faculty of Science, University of Helsinki, P.O. Box 64, Helsinki, 00014, Finland
[2]Extreme Environments Research Laboratory, Ecole Polytechnique Fédérale de Lausanne (EPFL), Sion, 1951, Switzerland
[3]Institute of Marine Science, Consejo Superior de Investigaciones Científicas (CSIC), Barcelona, Spain
[4]National Centre for Atmospheric Science, The School of Geography, Earth and Environmental Sciences, The University of Birmingham, Edgbaston, Birmingham, B15 2TT, United Kingdom
[5]Institute for Meteorology and Geophysics, University of Vienna
[6]Atmospheric Composition Research, Finnish Meteorological Institute, Helsinki, Finland
[7]Department of Environmental Science, iClimate, Arctic Research Center, Aarhus University, Roskilde, Denmark
[8]Environment and Climate Change Canada, Science and Technology Branch, Toronto, Canada
[9]Department of Applied Environmental Science (ITM), Stockholm University, 11418 Stockholm, Sweden
[10]Climate & Atmosphere Research Centre (CARE-C), The Cyprus Institute, P.O. Box 27456, Nicosia, 1645, Cyprus
[11]The National Research Centre for the Working Environment, Copenhagen, Denmark

*Correspondence to:* Matthew Boyer (matthew.boyer@helsinki.fi); Julia Schmale (julia.schmale@epfl.ch)

**Abstract.** The Arctic environment is rapidly changing due to accelerated warming in the region. The warming trend is driving a decline in sea ice extent, which thereby enhances feedback loops in the surface energy budget in the Arctic. Arctic aerosols play an important role in the radiative balance, and hence the climate response, in the region; yet direct observations of aerosols over the Arctic Ocean are limited. In this study, we investigate the annual cycle in the aerosol particle number size distribution (PNSD), particle number concentration (PNC), and black carbon (BC) mass concentration in the central Arctic during the Multidisciplinary drifting Observatory for the Study of Arctic Climate (MOSAiC) expedition. This is the first continuous, year-long dataset of aerosol PNSD ever collected over the sea ice in the central Arctic Ocean. We use a k-means cluster analysis, FLEXPART simulations, and inverse modeling to evaluate seasonal patterns and the influence of different source regions on the Arctic aerosol population. Furthermore, we compare the aerosol observations to land-based sites across the Arctic, using both long-term measurements and observations during the year of the MOSAiC expedition (2019 – 2020), to investigate interannual variability and to give context to the aerosol characteristics from within the central Arctic. Our analysis identifies that, overall, the central Arctic exhibits typical seasonal patterns of aerosols, including anthropogenic influence from



Arctic Haze in winter and secondary aerosol processes in summer. The seasonal pattern corresponds with the global radiation,
surface air temperature, and the timing of sea ice melting/freezing, which drives changes in transport patterns and secondary
aerosol processes. In winter, the Norilsk region in Russia/Siberia was the dominant source of Arctic Haze signal in the PNSD
and BC observations, which contributed to higher accumulation mode PNC and BC mass concentration in the central Arctic
than at land-based observatories. We also show that the wintertime Arctic Oscillation (AO) phenomenon, which was reported
to achieve a record-breaking positive phase during January – March 2020, explains the unusual timing and magnitude of Arctic
Haze across the Arctic region compared to longer-term observations. In summer, the PNC of nucleation and Aitken mode
aerosol is enhanced, but concentrations were notably lower in the central Arctic over the ice pack than at land-based sites
further south. The analysis presented herein provides a current snapshot of Arctic aerosol processes in an environment that is
characterized by rapid changes, which will be crucial for improving climate model predictions, understanding linkages between
different environmental processes, and investigating the impacts of climate change in future Arctic aerosol studies.

## 1 Introduction

Atmospheric aerosols influence Earth's surface energy budget and has two distinct effects on the climate system. The direct
effect describes the ability of aerosols themselves to scatter and absorb incoming solar radiation (Charlson et al., 1991), and
the indirect effect describes the ability of aerosols to act as cloud condensation nuclei (CCN) and ice nucleating particles (INP)
to form clouds, which are even more efficient at scattering radiation (Twomey et al., 1984). Additionally, clouds act as grey
bodies that re-emit longwave radiation, an important effect over high albedo surfaces such as in the Arctic. Aerosol-cloud
interactions comprise the largest source of uncertainty in our understanding of climate change globally (Boucher et al., 2013;
IPCC, 2021), and the Arctic environment has been shown to be very sensitive to radiative forcing from aerosols and clouds
(Sand et al., 2016), with large modeling uncertainties (Schmale et al., 2021). Thus, investigating climate-relevant aerosol
processes in the Arctic is crucial.


The Arctic climate is warming 2-3 times faster than the global average due to Arctic Amplification, which is caused by various
simultaneously acting feedback mechanisms such as the ice-albedo feedback and aerosol-cloud radiative forcing (Serreze and
Barry, 2011; AMAP, 2021). Globally, aerosol and cloud processes lead to a net cooling that opposes greenhouse gas warming
(e.g., IPCC, 2021), however, aerosol-cloud interactions in the Arctic can lead to a warming effect that is comparable in
magnitude to the warming associated with greenhouse gases, depending on the season (Lubin and Vogelmann, 2006). These
aerosol-cloud interactions are sensitive to the presence of aerosols, which are known to exhibit a strong seasonal dependence
in the Arctic (e.g., Lange et al., 2018; Schmale et al., 2022; Pernov et al., 2022).

Several key seasonal characteristics of the Arctic aerosol population have been previously identified. In winter, Arctic Haze,
i.e., the build-up of anthropogenic pollution from lower latitudes, is a frequent phenomenon over the Arctic region (Barrie,





1986; Leaitch et al., 1989). During the cold and dark winter months, low atmospheric moisture content and strong surface inversion layers limit aerosol depositional processes (Bradley et al., 1993; Shaw, 1995), and expansion of the polar dome permits transport from continental regions further south, particularly continental Europe and Asia (Eckhardt et al., 2003; Stohl 2006). Arctic Haze has been observed to occur during winter and persist through spring at several land-based sites across the Arctic (Heidam et al., 1999; Quinn et al., 2007; Tunved et al., 2013; Freud et al., 2017; Sharma et al., 2019). Previous observations of aerosol size distributions around the Arctic show that Arctic Haze is dominated by accumulation mode aerosol, or particles > 100 nm in diameter, while summer features a dominant Aitken mode and a Hoppel minimum, which indicates the strong effect of cloud processing on the aerosol population (Tunved et al., 2013; Croft et al., 2016; Asmi et al., 2016; Nguyen et al., 2016; Freud et al., 2017). At the same time, this demonstrates the importance of aerosol particles in cloud formation processes. Indeed, previous research suggests that the anthropogenic pollution associated with Arctic Haze can enhance longwave radiation emission from Arctic clouds, leading to surface warming (Garrett and Zhao; 2006).

In the transition from spring to summer, transport patterns and meteorological conditions change such that the advection of particulate pollution to the Arctic boundary layer from lower latitudes is limited (Stohl, 2006; Bozem et al., 2019). As a result, anthropogenic pollution is less common, and natural aerosol sources become more prevalent (Moschos et al., 2022a; Moschos et al., 2022b). Regional secondary aerosol formation, or new particle formation (NPF), occurs readily in summer due to enhanced biological activity and the resulting gas-phase emissions coupled with increased photochemistry (Nguyen et al., 2016; Burkart et al., 2017; Collins et al., 2017; Dall'Osto et al., 2018; Croft et al., 2019; Beck et al., 2021, Nøjgaard et al., 2022). In the central Arctic, the exact processes leading to new particle formation remain mostly unresolved (Croft et al., 2019; Schmale and Baccarini, 2021). There is evidence to suggest that cloud formation is limited by the availability of CCN in summer, as aerosols acting as CCN can be so sparse at times during this season that cloud formation is even inhibited (Mauritsen et al., 2011). Even a modest increase in aerosol or CCN concentrations in a clean environment, such as the summertime Arctic atmosphere, can have a significant impact on aerosol climate-relevant effects (Murphy et al., 1998; Carslaw et al., 2013; Karlsson et al., 2022).

Considering the accelerated warming in the Arctic, changes in natural aerosol are expected due to the decline in sea ice coverage, a larger extent of boreal forest fires, and changes in ocean biology in response to warming (Schmale et al., 2021 & 2022). These natural aerosol components, particularly organics, have been demonstrated to be as abundant in summer as anthropogenic organic components in winter (Moschos et al., 2022b). Hence, it is important to study the change in both anthropogenic and natural aerosol components throughout the year in the Arctic, whereby the latter are indicators of climate change feedback processes. However, despite this knowledge, there are few in-situ observations of Arctic aerosols, and many previous analyses are limited to short measurement periods (e.g., Chang et al., 2011), modeling studies (e.g., Croft et al., 2016), or land-based observations around the Arctic Ocean (e.g., Freud et al., 2017). There is a clear need for more comprehensive





studies to resolve the climate-relevant aerosol processes that occur in the Arctic atmosphere throughout the year and during

seasonal transitions (Schmale and Baccarini, 2021; Schmale et al., 2021).

To improve the understanding of aerosol processes in the central Arctic atmosphere, this study examines a continuous, year-long record of aerosol measurements obtained over the central Arctic Ocean during the Multidisciplinary drifting Observatory for the Study of Arctic Climate (MOSAiC) expedition (September 2019 – October 2020). This is the first continuous, year-

long dataset of aerosol PNSD ever collected over the sea ice in the central Arctic Ocean, which will be useful for evaluating the seasonality of aerosol properties over the sea ice. We use a targeted monthly approach to characterize the seasonal changes in the aerosol particle number size distribution (PNSD), particle number concentration (PNC), black carbon (BC) mass concentration, and air mass source regions throughout the year. In addition, we compare the MOSAiC aerosol observations to long-term data records from land-based sites across the Arctic to provide context on interannual variability during the year of

the MOSAiC campaign. This comparison allows us to gain new insights on aerosol properties across the Arctic region, including over the Arctic Ocean, during a full annual cycle. Furthermore, our analysis provides a snapshot of the processes that influence Arctic aerosols, which are subject to change in the rapidly warming Arctic. These results will be important for investigating the impacts of climate change in future Arctic aerosol studies.

## 2 Methods

The MOSAiC campaign was designed to address the scarcity of data available from the central Arctic. In fact, the MOSAiC expedition was the most comprehensive expedition to the central Arctic in history, and researchers evaluated the central Arctic environment for an entire year from various perspectives, including oceanography, sea ice dynamics, biology, meteorology, and atmospheric physics and chemistry. The possible insights from the MOSAiC expedition are numerous, as the collocated measurements of various Arctic system processes allow for interdisciplinary analyses of interlinked environmental processes

(Shupe et al., 2022; Nicolaus et al., 2022; Rabe et al., 2022).

### 2.1 The MOSAiC drift

The main premise of the MOSAiC expedition was to collect a year-long record of data while drifting in the Arctic sea ice on board the research vessel *Polarstern*. A map of the drift path is given in Fig. 1. The drift started in early October 2019 at 85°N, 136°E when *Polarstern* anchored to a suitable ice floe (Krumpen et al., 2020). Throughout the following year, the ship drifted

passively in the sea ice beside the selected ice floe, with a few exceptions. These exceptions include a short cruise to Svalbard, Norway for a crew exchange between mid-May and mid-June 2020 and a cruise to relocate the ship further north after the drift resulted in the ship entering the marginal ice zone at the end of July 2020. It is worth noting that the instruments used in this study were installed on *Polarstern* and were operational during these cruises as well, except when *Polarstern* was in the territorial waters of Svalbard between June 3 – June 8. The measurements for the campaign concluded at the end of September





2020. Apart from the crew exchange in Svalbard and a brief period when the ship entered the marginal ice zone at the end of
July, all these measurements were obtained in the central Arctic Ocean above 80°N within the pack ice, referred to hereafter
as the central Arctic.

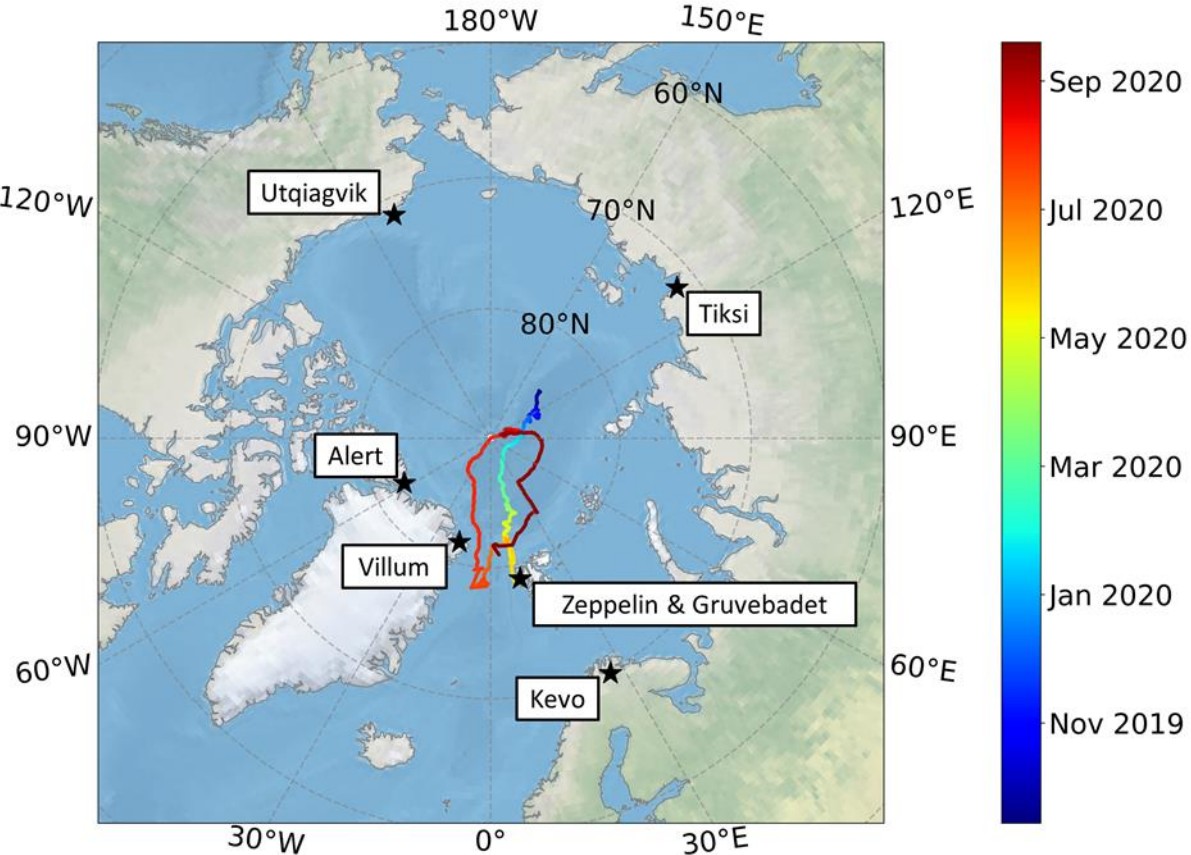

**Figure 1: The drift track during MOSAiC.** The color bar depicts the time of year. The black stars show the location of the land-based
sites across the Arctic that are referred to in this study. Note that Zeppelin and Gruvebadet are both located in Ny-Ålesund, Svalbard, Norway,
however, the two stations are located at different altitudes. Gruvebadet is ~50 m above sea level, whereas Zeppelin is ~500 m above sea
level.

**2.2 Aerosol size distribution and number concentration measurements**

In this study, a commercial Scanning Mobility Particle Sizer (SMPS) (TSI, inc.) was used to measure the ambient PNSD during
the MOSAiC campaign from 11 October 2019 to 1 October 2020. The ambient PNC was calculated by integrating over the
PNSD, which was evaluated in three size categories in this study: 10 – 25 nm (nucleation mode), 25 – 100 nm (Aitken mode),
and 100 - 500 nm (accumulation mode). The PNSD data was also used to calculate the condensation sink using the method
described in equations 5 & 6 in Kulmala et al. (2012).





The SMPS was installed in the Aerosol Observing System (AOS) container that was operated as part of the United States Department of Energy Atmospheric Radiation Measurement (ARM) user facility onboard *Polarstern*. The ARM AOS is a standardized measurement container that contains a suite of aerosol measurement instrumentation. The AOS used during the MOSAiC expedition was deployed as part of a larger ARM Mobile Facility (AMF) that was designed for ship-based deployments. The AOS was located on the port side of *Polarstern's* bow and was equipped with a total aerosol inlet that was

5 m in length, which corresponds to an inlet height of approximately 18 m above the sea surface. The internal temperature of the AOS container was maintained between 18-22°C during the campaign. A brief description of the SMPS is provided here; a more thorough explanation of the ARM SMPS instruments and their operating principles can be found in Kuang (2016), and for a detailed description of the other aerosol instrumentation in operation inside of the AOS container, as well as an overview of the AOS measurement objectives, design, and deployment history, refer to Uin et al. (2019).


   The SMPS consisted of an electrostatic classifier (TSI model 3082), a krypton aerosol neutralizer (TSI 3077a), a differential mobility analyzer (TSI model 3081), and a condensation particle counter (CPC) (TSI model 3772). An impactor with a diameter of 0.071 cm was placed in front of the inlet of the SMPS, corresponding to a cut-off diameter of ~720 nm. The ambient air sample was dried with a Nafion dryer (Perma Pure) prior to sampling. The SMPS was operated with an inlet flow

rate of 1 lpm and a sheath flow rate of 5 lpm. Under these conditions, the SMPS was able to perform 5-minute scans for particles between 10 – 500 nm in diameter. To verify and maintain the operation of the SMPS, weekly zero tests were performed with a high-efficiency particulate air (HEPA) filter placed on the instrument inlet, the inlet flow rate was measured, and the impactor was cleaned.

For PNC closure analysis (not shown here), the SMPS integrated PNC was compared to the particle concentrations reported from a collocated standalone CPC (TSI 3772). This CPC, called the CPCF (CPC Fine), was operated with a lower limit cutoff diameter, $D_{50}$, of 10 nm. The concentration closure analysis identified that SMPS agreed within 15% of CPC PNC for 85% of the time during the campaign. The concentration data from the CPCF was also used to create the pollution mask that was applied to the SMPS size distribution data, as described below in Section 2.4. For more details on the CPCF, refer to Kuang

(2016b).

   PNSD data from other land-based sites across the Arctic (see Fig. 1) were evaluated to compare with the MOSAiC observations. These land-based sites and the corresponding time ranges include Alert, Nunavut, Canada (2011–2018), Tiksi, Russia (2010–2018), Utqiagvik, Alaska, United States (2007–2015), Villum Research Station (Villum), Greenland, Denmark

(2010–2020), and Zeppelin, Svalbard, Norway (2010–2020). Information on measurement sites, instrumentation, processing, and quality control is available for Alert (Croft et al., 2016; Freud et al., 2017), Tiksi (Asmi et al., 2016), Utqiagvik (Kolesar et al., 2017), Villum (Nguyen et al., 2016), Zeppelin (Tunved et al., 2013).


### 2.3 Black Carbon

The black carbon (BC) mass concentration was measured using an aethalometer (AE33, Magee Scientific, Berkeley,
California, USA) at a wavelength of 880 nm. The instrument was installed in the *Swiss* container, which was located on the
starboard side of the *Polarstern*'s bow, directly adjacent to the AOS container (refer to Fig. 3 in Shupe et al., 2022). The
aethalometer sampled behind an automated valve, which switched hourly between a whole air inlet and an interstitial inlet
with an upper cut-off size of 1 μm (Beck et al., 2022). The inlet flow was set to 2 lpm and was verified biweekly. The instrument
collected mass concentration data at a time resolution of one second, which was then averaged to 10 minutes for further
analysis. Outliers in the BC data of more than 3 times the standard deviation on an hourly moving window were removed. We
used the standard mass absorption cross-section value of 7.77 $m^2$ $g^{-1}$ and no Cref value (Drinovec et al., 2015). Note, we use
the abbreviation BC throughout the manuscript, while this measurement represents equivalent BC (eBC).

BC and EC mass concentrations from other land-based sites across the Arctic spanning several decades are included for a
comparison with the BC measurements from MOSAiC. These land-based sites and the corresponding time ranges include
Alert, Nunavut, Canada (1989–2017, Aethalometer), Gruvebadet, Svalbard (2010–2018, PSAP,
https://data.iadc.cnr.it/erddap/tabledap/ebc_2010_2020.html), Kevo, Finland (1964–2010, elemental carbon, thermal optical
method), Tiksi, Russia (2009–2018, Aethalometer), Utqiagvik, Alaska, United States (1991–2019, PSAP, CLAP,
Aethalometer), Villum, Greenland (2009–2018, elemental carbon, thermal optical method), and Zeppelin, Svalbard, Norway
(2001–2017, aethalometer). eBC data from Villum, Greenland between 2018 – 2020 were obtained using an Aethalometer, as
detailed in Thomas et al. (2022).  For details on the measurements of the eBC and elemental carbon mass, we refer to Schmale
et al. (2022).

### 2.4 Pollution masking

One of the challenges associated with ship-based campaigns is contamination of the ambient sample by emissions from the
ship stack or other local pollution sources (Beck et al., 2022). A pollution detection algorithm was applied to the CPCF particle
concentration data, which was collocated in the ARM AOS container, to identify and remove pollution. The details of the
pollution detection algorithm are discussed in Beck et al. (2022). As input parameters for the pollution detection algorithm,
threshold parameters of m=0.64 and a=0.5 were used for the derivative filter, and a median time of 60 minutes was used with
an interquartile range threshold of 1.3 (see Beck et al. (2022) for details on the parameters). The pollution mask for the CPCF
was calculated at a time resolution of 1 minute. The CPCF pollution mask was then used to remove scans from the SMPS
influenced by pollution. If any minute during the 5-minute SMPS scans was flagged as polluted according to the CPCF mask,
the entire 5-minute scan was discarded. 42% of the annual aerosol size distribution dataset remained after the application of
the pollution detection algorithm. Refer to Fig. S1 in the Supplementary Information (SI) for an example of the pollution





detection algorithm's use on the SMPS data and Fig. S2 for an overview of the data coverage during each month after pollution
removal.

The BC mass concentrations were also masked using the pollution masking algorithm developed by Beck et al. (2022). Particle
concentrations obtained from a CPC (TSI, model 3025) which was collocated with the BC instrument inside the *Swiss*
Container were used for masking the BC data. Note that two different CPCs were used for pollution masking in various
instruments due to the differences in instrument location during the campaign. The choice of CPC used for pollution detection
corresponds with the location of the instruments in different measurement containers (the ARM AOS and *Swiss* container)
with differing inlets such that the pollution masks are more representative of the measurements from the specific inlets of the
two different containers.

**2.5 FLEXPART simulations**

Backward simulations with the Lagrangian particle dispersion model FLEXPART v10.4 (Pisso et al., 2019) were performed
to determine the origins of the prevailing air masses and to evaluate the contribution from different source regions. The model
has been driven with hourly meteorological data from the ERA5 reanalysis with 0.5° x 0.5° resolution. A cluster of 100,000
atmospheric particles was initialized every three hours along the ship track and traced backward in time up to 30 days. The
simulations were computed for a passive air tracer without removal processes as well as for a black carbon (BC) tracer with
wet and dry removal. The main output from FLEXPART consists of 3-dimensional fields of emission sensitivity, sometimes
also referred to as "source-receptor relationship" (SRR) (e.g., Seibert and Frank, 2004), which describes the influence that a
unit emission flux intercepted by atmospheric transport would have on a tracer concentration at the ship's location. We will
focus for this paper on the emission sensitivity close to the surface (below 100 m, the lowest model output layer), also referred
to as "Footprint Emission Sensitivity" (FES), as most emissions of the tracers that we are interested in occur at the surface.
When multiplying the FES with emission flux data from an emission inventory, we also obtain source contribution maps (e.g.,
Fig. S11), and by spatial integration of the source contributions, we obtain the simulated concentration at the ship location
(e.g., Fig. 8).

**2.6 Source region identification using an inverse model**

Inverse modeling (Seibert, 1998) was used in this study to identify—and derive the strengths of—potential source regions of
emissions or their precursors based on in situ measured concentrations and the SRR matrix produced by the FLEXPART
dispersion model simulations. The potential sources are identified by solving Eq. (1):

$$y = Ax + n ,$$
(1)

where y is the measurement vector, x is the source term, A is the transport matrix—here the surface FLEXPART SRR—and
n is the error.




Solving the equation for x is not trivial, and in this study, we overcome the difficulty by (1) reducing the dimensionality of the A matrix to 200 groups by clustering its cells based on the time series of their influence (Aliaga et al., 2021; Faletto and Bien, 2022), and (2) by imposing an elastic net regularization (similar methods are proposed by Martinez-Camara et al. (2014) and references therein) with iteratively relaxing constraints  (36 iterations for the hyperparameters). The result of this procedure

creates a set of 36 source region foot-print maps (e.g., Fig. S10) from which we chose an iteration that compromises between capturing most of the a priori known source areas while ignoring noisy regions. Finally, we can obtain a time series of potential influence from the identified source region polygons to provide insight into seasonal variation in the source regions.

**2.7 Aerosol size distribution cluster analysis**

The PNSD measurements provided by the SMPS from MOSAiC were aggregated into hourly (4914 for MOSAiC) and daily

(297) arithmetic means. The daily PNSD measurements were then normalized to their vector length, and a cluster analysis was performed using k-means according to the Hartigan-Wong method (Hartigan and Wong, 1979), which is an established method for evaluating aerosol size distribution characteristics (Beddows et al., 2009; Beddows et al., 2014; Freud et al., 2017, Pernov et al. 2022). By normalizing the data to the vector length, the shape of the PNSD, rather than its magnitude, was clustered. Since the cluster analysis was performed on the daily PNSD measurements, the resulting clusters show the typical PNSD for

a day, where each day of PNSD data is assigned to a single cluster. The cluster analysis was carried out using 2 to 10 clusters, of which the 9-cluster output best described the data. The optimum number of clusters (9) was derived using the Dunn Index and Silhouette Width, as previously described in similar studies (Dall'Osto et al., 2017; 2019). The results of the PNSD cluster analysis are described in Sections 3.2 and 3.3.

**2.8 Ancillary atmospheric measurements**

Environmental variables of ancillary atmospheric measurements are included to provide context for the conditions in the ambient atmosphere during the MOSAiC campaign. These measurements include ambient air temperature and global radiation, or short-wave downwelling radiation. Ambient air temperature was measured using a Vaisala HMP155 located 29 m above the sea surface. Global radiation was measured using a Pyranometer (Kipp & Zonen, CM11) installed 34 m above the surface. Both sensors are part of the meteorological observatory installed onboard *Polarstern* that operates continuously during ship

operation. Operation of the sensors was checked daily throughout the campaign, and the datasets were quality controlled to remove erroneous data points. Please see the section on data availability for DOIs to the specific datasets used herein.

**3 Results and discussion**

**3.1 FLEXPART source region analysis**

Figure 2 presents the FES of various source regions, including Europe, ocean, sea ice, North Asia, and North America,

throughout the MOSAiC campaign using the FLEXPART simulation data. We present this analysis first, as it provides useful





information for the aerosol measurements discussed in later sections. Overall, the ocean and sea ice regions had the most prevalent surface influence during the year. There is a clear seasonal cycle in the surface influence from the other source regions (Europe, North Asia, and North America). Most notably, these continental regions further south had relatively higher FES values from November - April, especially from North Asia. Much lower contributions from the continental regions were observed during May – July. The results then show an increase in the surface influence from the continental regions towards the end of the campaign in August and September.

These FES results are as expected and agree with previous analyses of air mass transport patterns in the Arctic. Transport times from southerly source regions in the central Arctic are substantially longer in summer than winter (Stohl, 2006). During summer, a shrinking effect of the polar dome, or a meteorological boundary region that varies in time and space around the Arctic, inhibits air mass advection, such that the horizontal transport of aerosol particles northward at the surface of the central Arctic is limited by regional meteorological characteristics (Bozem et al., 2019). The continental regions further south are characterized by anthropogenic influence, and hence the seasonality of these transport patterns is also consistent with previous observations of Arctic Haze (Tunved et al., 2013), as discussed in Sections 3.2.1 and 3.2.2 below. The enhancement in the surface contributions from continental regions in August and September may be explained according to the location of *Polarstern* during MOSAiC. In August and September, *Polarstern* transited further north and into the eastern sector of the central Arctic (Fig. 1), which may impact the influence of different source regions compared to the ship's location earlier in the summer due to changes in transport patterns between the two regions (Stohl, 2006).





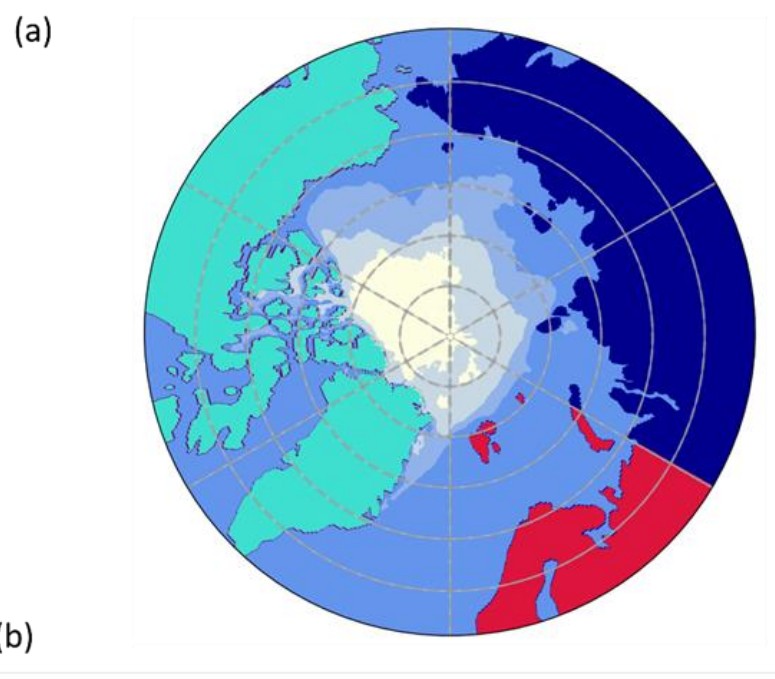

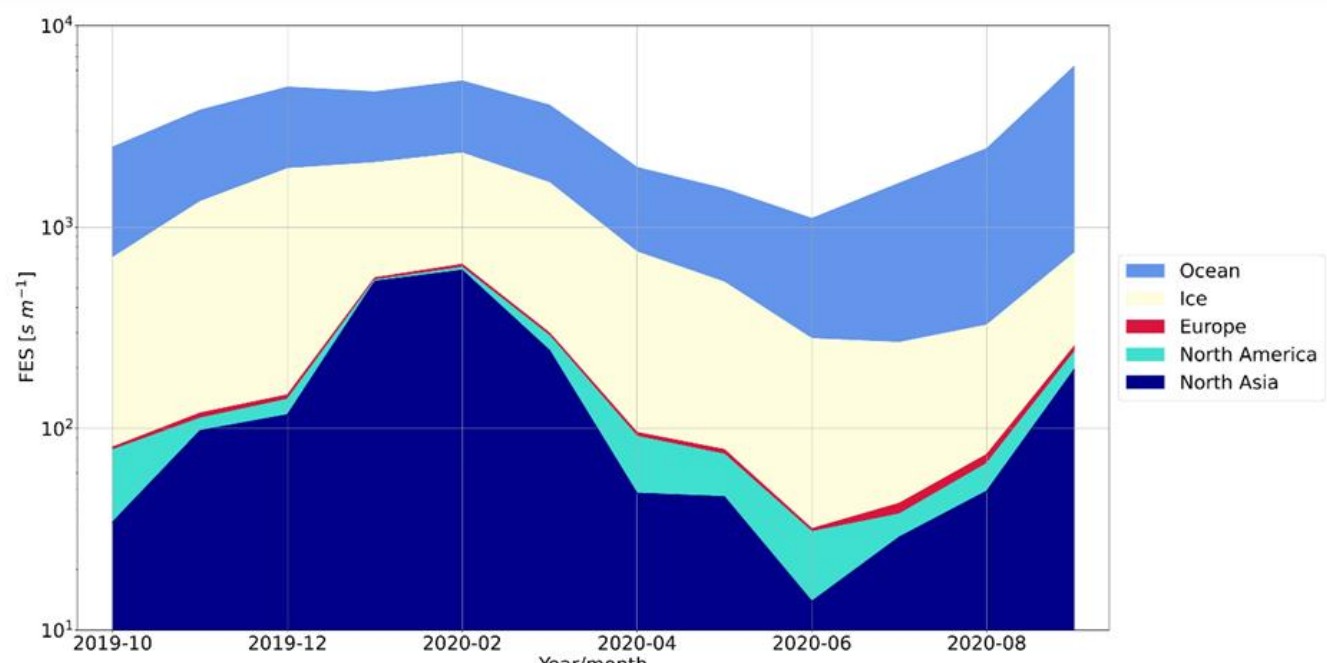

**Figure 2: The seasonality of surface influence from air mass source regions using FLEXPART.** The FLEXPART air tracer data within the lowest 100 meters of the atmosphere, or the FES, was used to identify the influence of different source regions on the observed aerosol throughout the year. The FES was quantified for each source region by applying a geographical mask (a) to the FLEXPART air tracer data.





## 3.2 k-means aerosol size distribution cluster analysis results

As previously mentioned, the k-means cluster output identified nine aerosol clusters, however, similar cluster types
(accumulation mode and nucleation mode clusters) were further grouped manually to simplify interpretation. The result of the
cluster analysis after manual grouping yields five clusters that broadly describe the Arctic aerosol population, as shown in Fig.
3. Each cluster is presented as the average of all days associated with each cluster type, and the clusters were named according
to the characteristics of the average daily distributions. Refer to Fig. S3 for the average distributions of the nine aerosol clusters

before manual grouping and to Fig. S4 for disaggregated hourly averages of the nine daily PNSD clusters. The grouped clusters
include a nucleation mode cluster, an Aitken mode cluster, an accumulation mode cluster, a bimodal cluster, and a clean cluster.
In the following, a brief description of the cluster characteristics is given. The nucleation mode cluster has peak concentrations
below 25 nm, indicative of secondary aerosol processes and growth (Kecorius et al., 2019). The Aitken mode cluster has a
diameter mode at 46.1 nm, which suggests aging of new particles (Lawler et al., 2021). The combined accumulation mode

cluster is characterized by an average mode at 174.5 nm (modes of 181.1, 224.7, 174.7, and 117.6 nm for all accumulation
mode clusters, see Fig. S3), and likely represents aerosols of anthropogenic origin (Lange et al., 2018). The bimodal cluster
has a peak in both the Aitken and accumulation modes, with modal diameters of 46.1 and 135.8 nm, respectively, indicating
that this cluster has possible contributions from natural as well as anthropogenic sources. The clean cluster has the lowest
overall concentration compared to the other clusters and features two modal features that correspond to the nucleation and

accumulation modes, which again suggests possible influence from natural and anthropogenic sources. These clusters are
generally similar to previous cluster results from other Arctic stations (Dall'Osto et al., 2017; Pernov et al., 2022). The
differences mainly arise from different periods, different size ranges, different temporal aggregation, and the fact that land-
based stations are stationary while *Polarstern* drifted in the central Arctic Ocean. The occurrence of each cluster throughout
the year is presented in the following section to supplement the analysis of the annual cycle in the PNSD.




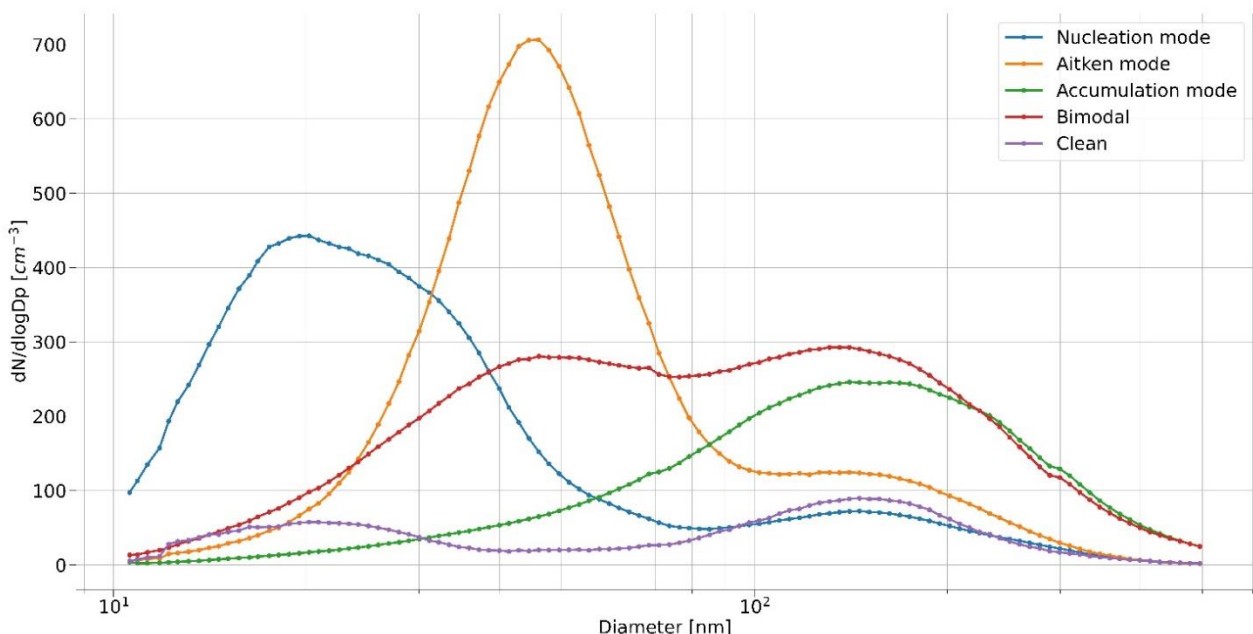

**Figure 3: The average distributions of the PNSD clusters.** The particle concentration in units of dN/dlogD$_P$ is presented for the average of all days in each cluster according to the particle size. Note that the nucleation and accumulation mode clusters presented here are the result of manually grouping the two nucleation mode clusters and four accumulation mode clusters into a single nucleation and accumulation mode cluster, respectively. Refer to Fig. S3 for the original 9 cluster output.

## 3.3 Annual cycle of the aerosol size distribution

The annual cycle of the PNSD collected during MOSAiC is presented as monthly median distributions together with the monthly air temperature in Fig. 4, and the fraction of occurrence of each grouped PNSD cluster is shown in Fig. 5. For further context, the annual cycle in the PNC according to different size modes is given in Fig. 6. Refer to Fig. S5 in the SI for the PNSD plot in daily resolution. Monthly representations of the size distributions are shown in Section 3.5 with the comparison to land-based aerosol observations. In general, the most pronounced differences in the PNSD occur between winter and summer, which can be explained according to seasonal transitions in the meteorology and atmospheric dynamics of the Arctic. In this section, we present and discuss the changes in the PNSD according to the season.



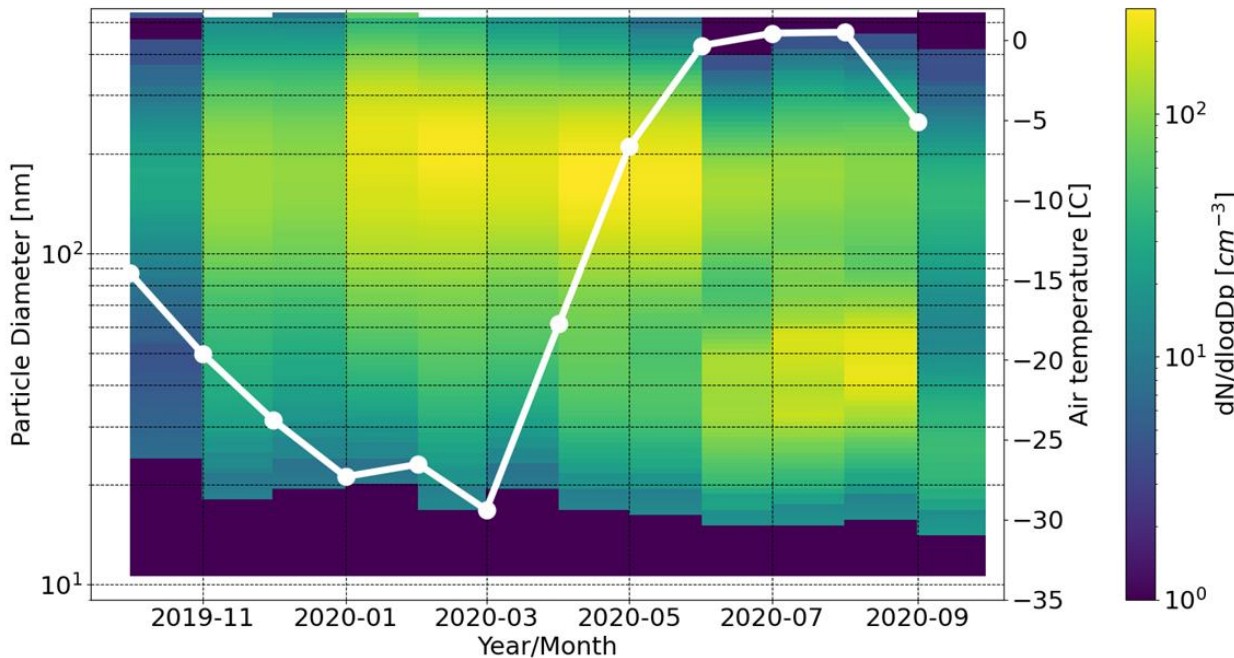

**Figure 4: The annual aerosol size distribution and air temperature in the central Arctic during MOSAiC.** The size distribution data are presented as monthly medians of the PNSD bins, where the concentration of the bins is represented by the color bar in units of dN/dlogD$_p$. The surface air temperature (in white), depicted as monthly averages, is included to give context to seasonal changes in the aerosol size distribution.





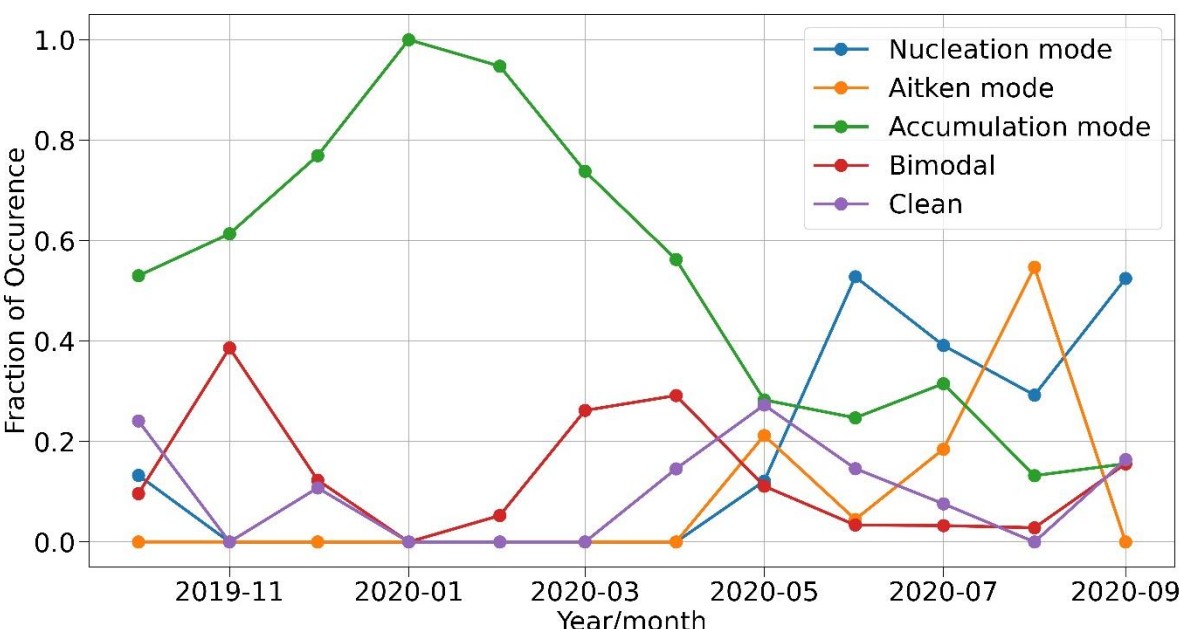

**Figure 5: The monthly fraction of occurrence for the PNSD clusters.** The fraction of occurrence shows the total number of days per
month that each cluster is represented compared to all other PNSD clusters.

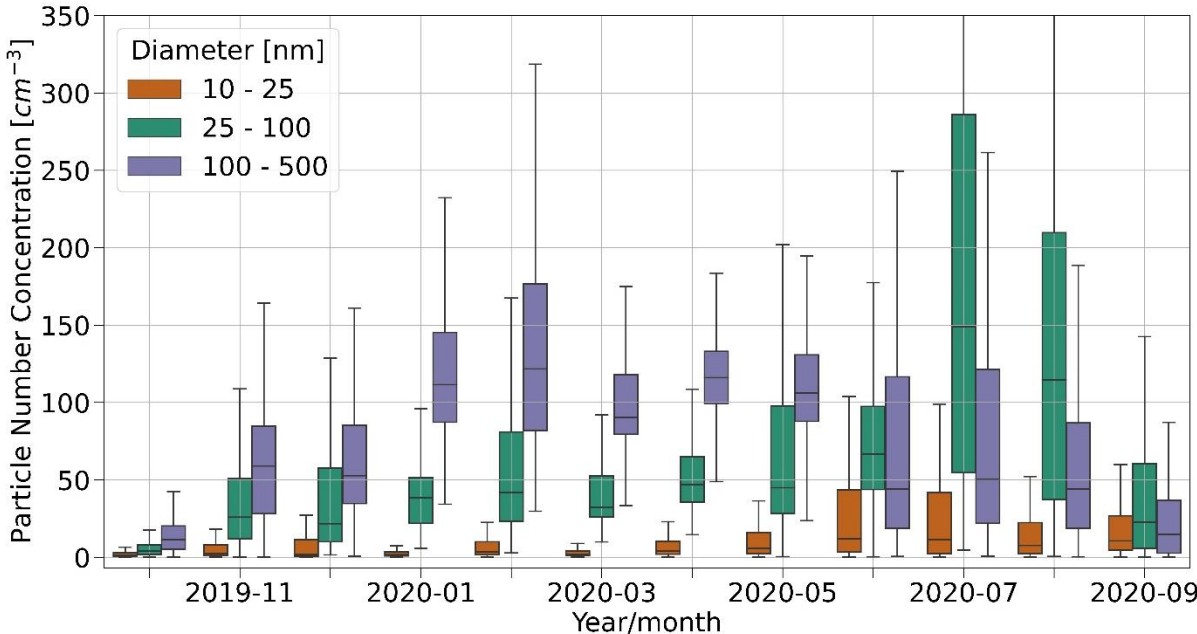

**Figure 6: The annual cycle in the PNC during MOSAiC in different size ranges.** The box plots represent the PNC as monthly medians
of the integrated SMPS data after the removal of local pollution sources. The size ranges are defined as 10–25 nm (nucleation mode, orange),
25–100 nm (Aitken mode, green), and >100 nm (accumulation mode, purple).






### 3.3.1 Winter

Winter in the Arctic (December – February) is characterized by complete darkness, cold surface temperatures (average air temperature during MOSAiC between December 2019 – February 2020 = -26 °C, std = 5 °C), and relatively stable sea ice conditions under a strong, dry surface atmospheric inversion layer typical for that season (Bradley et al., 1993). The dry and

stable atmospheric conditions create a situation where horizontal transport of aerosol from lower latitudes can occur due to limited wet deposition processes (Klonecki et al., 2003), and winter experiences a higher prevalence of air masses from southerly, continental source regions, as previously discussed (Section 3.1, Fig. 2). These advected air masses arriving in the central Arctic are often influenced by anthropogenic particulate pollution episodes, especially during the winter season, commonly known as Arctic Haze (Quinn et al., 2007). The Arctic Haze season can extend also beyond the winter months,

from late fall through spring. As a result, the PNSD observed during the MOSAiC expedition is dominated by accumulation mode particles with an average modal diameter of 186 nm (std = 23 nm) from November 2019 to May 2020. The k-means cluster analysis of the PNSD data, as shown in Fig. 5, identifies that accumulation mode clusters occur 70% (std = 24%) of the time during this period (November to May) on average, with a maximum during January (100%) and a minimum during May (28%). The average monthly accumulation mode PNC (particle diameters > 100 nm) between November – May is 92

$cm^{-3}$ (std = 27 $cm^{-3}$), and we observed a maximum accumulation mode concentration of ~120 $cm^{-3}$ during February. Other studies have observed that Arctic Haze peaks in April (Tunved et al., 2013; Freud et al., 2017), however, in this dataset, there does not appear to be a significant change in the accumulation mode PNC between January – May (avg = 108 $cm^{-3}$, std = 9 $cm^{-3}$). It is also noteworthy that the concentration of accumulation mode particles is less pronounced during late fall/early winter after the sun has set and increases as winter progresses (Figs. 5 & 6). More specifically, the accumulation mode PNC

was ~50 $cm^{-3}$ in November and December, followed by stable concentrations twice the magnitude from January – May.

The FLEXPART FES analysis also identifies that the continental source regions, particularly North Asia, have a distinct influence during this time of year (Fig. 2). The influence from North Asia is especially pronounced during January and February. These regions are characterized by anthropogenic activities, and combined with the favorable transport conditions

during winter, this observation is consistent with the highest accumulation mode PNC observed during February.

### 3.3.2 Spring

Arctic haze persists during spring (March – May) because the conditions remain favorable for air mass transport from lower latitudes, as previously discussed. Previous research has identified that air mass transport into the Arctic from lower latitudes is more favorable in winter and spring than in summer due to regional atmospheric conditions, especially cold surface

temperatures (Stohl., 2006; Bozem et al., 2019). Our observations of ambient air temperature support these previous studies; the mean monthly temperatures show a distinct pattern regarding the presence of Arctic Haze in the PNSD (Fig. 4). When the





surface temperatures are still significantly below freezing, including the period during the spring months after polar sunrise (see Fig. S6 for the annual cycle of global radiation and Fig. S7 for ambient air temperature in native resolution), we continue to see the occurrence of Arctic Haze. These cold air masses during this time of year have low moisture content, and the stability
of the atmosphere remains intact due to such low temperatures, which limits both wet and dry depositional processes (Shaw, 1995; Garrett et al., 2011). While we do not assess precipitation during transport in this study, others have shown that precipitation amounts are higher in summer than in winter/spring during air mass transport from lower latitudes to the Arctic (Barrie, 1986; Freud et al., 2017; Pernov et al., 2022). Therefore, transport of anthropogenic emissions from lower latitudes is possible, as particulate pollution has longer lifetimes under cold, dry, and stable conditions (Bradley et al., 1993). It is important
to note that our observations reflect the aerosol measured within the boundary layer at the surface of the central Arctic and do not represent processes occurring aloft in the atmosphere. However, recent studies have shown that Arctic Haze aerosol originating from biomass burning and anthropogenic sources can also be transported aloft over the central Arctic Ocean during spring (Quennehen et al., 2012; Ohneiser et al., 2021; Engelmann et al., 2021).

May itself represents the time when the Arctic aerosol processes undergo a seasonal transition; the occurrence of the accumulation mode from Arctic Haze subsides, and the nucleation and clean PNSD clusters become more prevalent during the month (Fig. 5). As the Arctic environment continues to change, these temperature-dependent transport processes may influence the future seasonality and prevalence of anthropogenic pollution as well as the transition to natural aerosols in the central Arctic (Browse et al., 2012), which can yield important implications for radiative processes in the Arctic atmosphere
(Shindell and Faluvegi, 2009).

### 3.3.3 Summer

The summertime central Arctic (June – August) experiences warmer surface temperatures than the rest of the year (average between June – August 2020 = 0.2 °C, std = 1.5 °C). When the average monthly temperatures rise above freezing at the end of May (Fig. 4) and the entire northern hemisphere has warmed, instability in the atmosphere is more pronounced and
convection readily occurs in air masses that are in transit to the central Arctic. Enhanced convection, combined with higher atmospheric moisture, has been observed to lead to efficient wet deposition processes, which serves as an efficient sink for transported aerosol particles (Browse et al., 2012). Thus, the conditions become less favorable for horizontal advection of particulate pollution that originates over regions with anthropogenic emissions. This effect was also demonstrated by Freud et al. (2017) who observed that enhanced precipitation along air mass trajectories leads to efficient aerosol removal in the Arctic
during summer. In addition, air mass transport from source regions characterized by anthropogenic influence reduces during summer, as discussed in Section 3.1 (Fig. 2).

The results of reduced aerosol transport to the Arctic are two-fold; there is a lower condensation sink, or a lower loss rate of condensable vapors onto pre-existing aerosol surfaces (Fig. S8), and natural aerosols sourced from regional processes within



the Arctic become more prominent. Therefore, particles sourced from secondary processes (in both the nucleation mode and Aitken mode) dominate the PNSD within the central Arctic boundary layer during summer, which is clearly visible in the marked changes in the PNSD (Fig. 4) and the PNSD cluster analysis (Fig. 5) from June through August 2020. The nucleation mode cluster is most prevalent in June and July and accounts for 40% (std = 11%) of the days between June – August on average. The summer months are also characterized by lower concentrations of accumulation mode aerosol and an increase in

Aitken mode particles, i.e., particles between 25 – 100 nm in diameter (Fig. 6), which are relevant for CCN formation (Leaitch et al., 2016). The Aitken mode cluster peaks at 55% during August. In addition, we see the highest total PNC of the year during the summer months, with a maximum monthly median PNC >250 cm$^{-3}$ in July. These observations agree with other studies that have identified the onset of NPF in Svalbard during late/spring and summer due to enhanced biological activity (Tunved et al., 2013; Dall'Osto et al., 2019; Lee et al., 2020) and frequent NPF and growth of Arctic aerosol in summer that may

contribute to CCN (e.g., Collins et al., 2017; Chang et al., 2022; Beck et al., 2021). Note that the annual cycle of precursor gases to aerosol formation will be presented in another paper, and the mechanism of NPF during the year will be evaluated in a separate study.

Another summertime feature of the monthly PNSD is the presence of a "Hoppel minimum" (Hoppel, 1994), defined as a

minimum concentration between the Aitken and accumulation modes in the size distribution from July through August (Fig. 4), which suggests that cloud processing has had an influence on the measured PNSD. The "Hoppel minimum," which is well known to describe cloud processing in the marine boundary layer (Hoppel 1994; Zheng et al., 2021), is centered around ~90 nm (std = 13 nm) in this dataset. It is important to note that cloud processing can have an important role on the aerosol and CCN populations (Hatzianastassiou et al., 1998; Flossmann and Wobrock, 2019; Karlsson et al., 2022). We do not evaluate

the impact of cloud processing further in this work, however, a total and interstitial inlet system capable of resolving cloud processing was operated in the *Swiss* container during MOSAiC (Beck et al., 2022). Future work will be carried out to evaluate the influence of cloud droplet residuals on the PNSD in the central Arctic during MOSAiC.

### 3.3.4 Fall

Global radiation in the central Arctic decreases rapidly in fall (September – November) (Fig. S6). The surface temperature

begins to drop and onsets freezing in the sea ice, which will remain frozen throughout winter. We observed that the freezing onset occurred between late August and early September, which corresponds to the drop in the average monthly air temperatures below the freezing point (-1.8°C for saltwater) in September (Fig. 4). Interestingly, we still see the dominance of NPF processes in the PNSD in September despite the onset of freezing, as the nucleation mode remains the dominant aerosol cluster, accounting for up to 52% of the days during that month (Fig. 5). This observation agrees with the results of Baccarini

et al. (2020), who observed NPF from iodic acid during the sea ice refreezing period that coincides with the observations of enhanced iodic acid concentrations during the year, which will be discussed in a separate study on precursor gases to aerosol





formation. Pernov et al. (2022) also observed a nucleation cluster to occur most frequently during September at Villum, which suggests that NPF occurs across the entire Arctic region during the freezing onset.

There is also a clear decrease in the PNC that occurs in mid-fall after the freezing onset and drop in temperature. The PNC falls to the lowest observed values of the year in October, with a monthly median < 20 cm$^{-3}$ (Fig. 6). The low concentrations can be attributed to the decrease in solar radiation and biological activity from the marine ecosystem, and hence lower concentrations of gas-phase aerosol precursors and therefore less nucleation and Aitken mode particles sourced from secondary aerosol formation processes. Furthermore, previous studies have identified that meteorological conditions in fall are not yet

favorable for the advection of aerosols from lower latitudes, combined with enhanced wet depositional processes during transport, yielding low aerosol concentrations in the Arctic (Croft et al., 2016; Freud et al., 2017). Our observations support these previous results and provide further observational evidence of these aerosol processes during the seasonal transition to winter.

**3.4 Annual cycle of black carbon**

Figure 7 presents the annual cycle of BC mass concentrations measured during MOSAiC (black solid line) together with the inverse modeling results. BC mass concentrations from MOSAiC reach an annual maximum between January – May 2020, with an average BC mass concentration of 71.0 ng m$^{-3}$ (std = 34.1 ng m$^{-3}$) during this time. Notably, the highest BC concentrations were observed in January and February (109.2 and 106.7 ng m$^{-3}$, respectively). Then, we observed much lower BC concentrations from June – October (average = 8.3 ng m$^{-3}$, std = 6.0 ng m$^{-3}$), followed by increasing concentrations again

in November and December. For BC mass concentrations at a 10 min resolution, see Fig. S9 in the SI.

The timing and magnitude of the BC mass concentrations provide further context to the PNSD and source region observations. The annual trend in the BC data is, in general, consistent with the timing of Arctic Haze in the PNSD and air mass transport from southern regions, as previously discussed. Notably, the BC mass concentrations correspond with the increased source

contributions from North Asia (Fig. 2) and the highest incidence of accumulation mode aerosol in the PNSD in January and February (Figs. 4, 5, & 6).





**Figure 7: Annual cycle of BC mass concentrations and inverse model results**. (a) The source contribution map for BC as simulated by the inverse model. The red color bar shows the annual average source contribution according to geographic location, determined using the FLEXPART air tracer data and the measured BC mass concentrations. The polygons on the map indicate the main source region clusters during the year, which were used to evaluate the simulated BC mass concentration time series. Polygon a, b, c, and d are generally characterized as Russia/Siberia, Europe, Alaska, and Greenland/Iceland, respectively. (b) The time series of source region cluster contributions to the simulated BC mass concentrations throughout the year. The BC mass concentrations measured during MOSAiC are included for reference, where the grey shaded region shows the interquartile range.







The inverse model was used to further identify the source regions contributing to BC during MOSAiC. The BC source region contribution map and simulated BC time series from the inverse model are presented in Fig 7. Refer to Fig. S10 for the maps showing the iterations of the hyperparameters using the elastic net regularization method used in the inverse model. According to the inverse model, source region polygon a (blue), which describes Russia/Siberia, is identified as the most significant

contributor to BC mass during MOSAiC, especially during January and February. This region in Russia/Siberia encompasses Norilsk, which is known as a large source of anthropogenic pollution due to the prevalence of smelters. This outcome further agrees with the FLEXPART FES analysis, as well as the PNSD observations. A similar approach was used with the FLEXPART FES data and simulated anthropogenic BC emissions from the ECLIPSE v6b emission inventory, and the results agree very well with results from the inverse model, as shown in Fig. 8. The corresponding source contribution maps, showing

the specific spatial distribution of the BC sources in the emission inventory, are given in Fig. S11. The results in Fig. 8 identify North Asia as the dominant anthropogenic source region of BC during the year, with a peak in BC mass during January. This is consistent with the inverse model and further highlights the influence of anthropogenic pollution from North Asia in the central Arctic during winter. In addition, the emission inventory source contribution maps (Fig. S11) show very similar trends in the spatial distribution of BC sources compared to the source regions identified by the inverse model, which provides further

validation of these results.

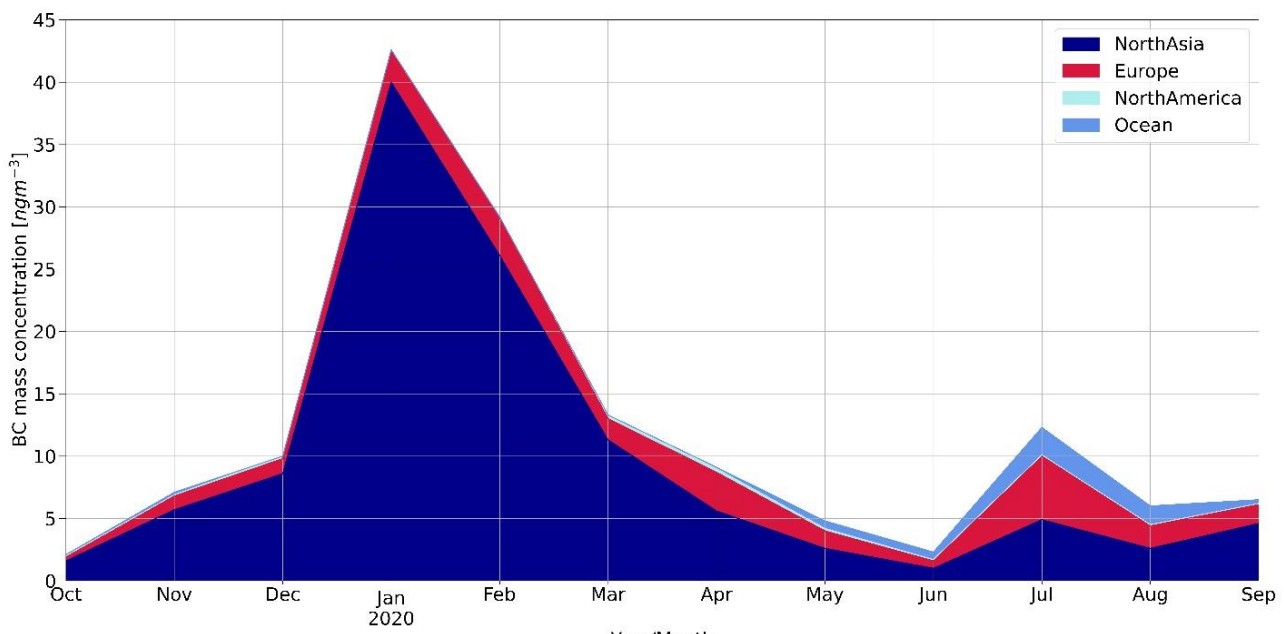

**Figure 8: Simulated source regions of anthropogenic BC concentrations from the ECLIPSE emission inventory.** The BC mass concentrations are presented as monthly averages for each of the source regions using the FLEXPART simulations and the ECLIPSE v6b emission inventory. The source region definitions correspond to the geographical source region mask as shown in Fig. 2.



### 3.5 Comparing MOSAiC observations to land-based sites

To provide additional context to the MOSAiC aerosol data, we compared the PNSD, PNC, and BC data to several land-based sites across the Arctic (see Fig. 1 for the site locations). There were two complementary approaches for this comparison: (1) a comparison of MOSAiC with long-term measurements from the land-based sites spanning several decades and (2) a comparison between MOSAiC and the sites where data was available during 2019 – 2020 (Villum and Zeppelin). Since the MOSAiC data is limited to a single year, these two comparison approaches provide insight into the effects of interannual variability on the observations during the year of the MOSAiC campaign.

#### 3.5.1 Comparison to long-term aerosol observations

Figure 9 presents the comparison of MOSAiC PNSD and PNC measurements with the long-term aerosol observations from the land-based sites, and Figure 10 shows a similar comparison for BC mass concentrations. Figure 9a shows similar features in the annual cycle of the PNSD at all land-based sites, including the buildup of Arctic Haze during winter and spring, secondary aerosol formation processes during summer contributing to particles in the smaller size bins, and the lowest aerosol concentrations of the year in the fall. Similar trends are also apparent in the PNC data for all sites (Fig. 9b), where the highest concentrations of smaller particles (in both 10 – 25 and 25 – 100 nm size ranges) are observed in summer and the accumulation mode particles (> 100 nm) dominate in the winter/spring. The same seasonal trend of Arctic Haze is observed in the BC data at all sites (Fig. 10). Such a result is not surprising, as aerosol observations from these sites have already been evaluated in detail (Freud et al., 2017; Schmale et al., 2022). The largest differences in the long-term observations among the land-based sites occur in Tiksi and Utqiagvik, which typically have more variation in the shape of the PNSD and a higher PNC throughout the year. These differences are likely due to the proximity of these sites to large emission sources, and as a result, the data from Tiksi and Utqiagvik may not be as representative of the baseline aerosol concentrations across the larger Arctic region as the other sites (Asmi et al., 2016; Gunsch et al., 2017; Freud et al., 2017; Popovicheva et al., 2019).





(a)

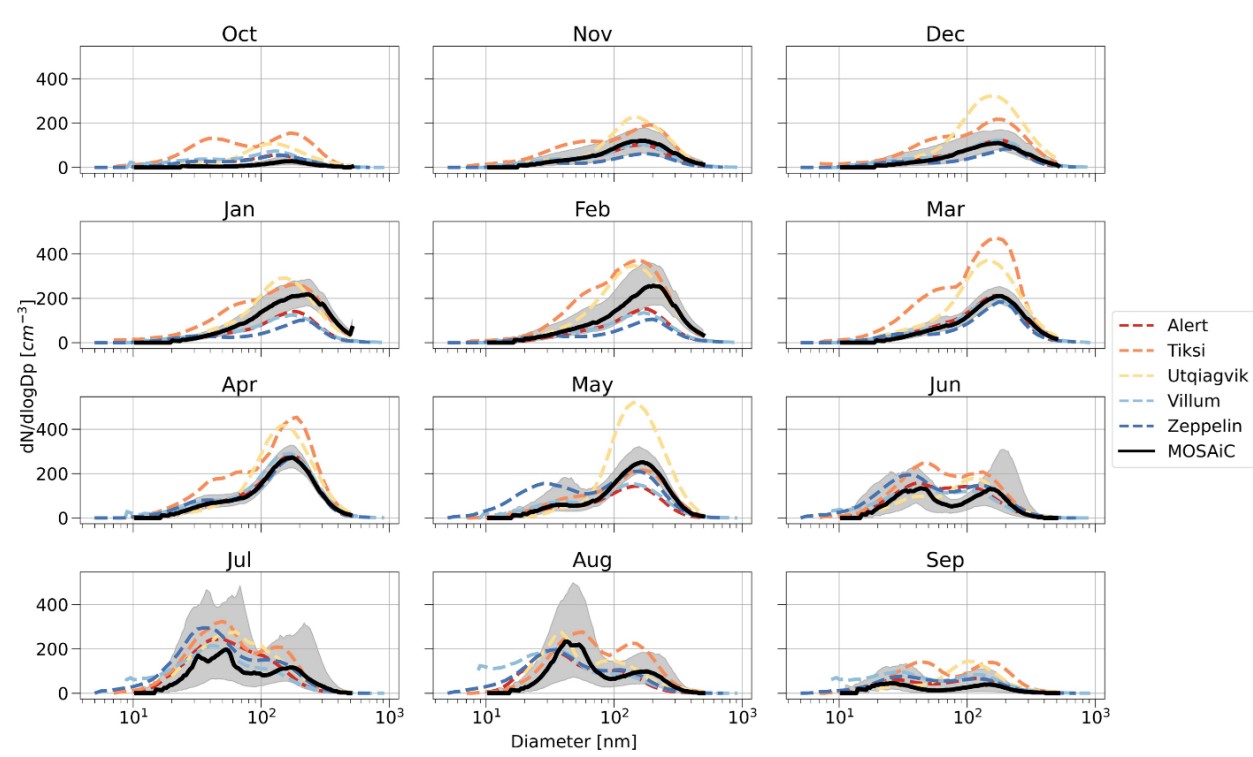

(b)

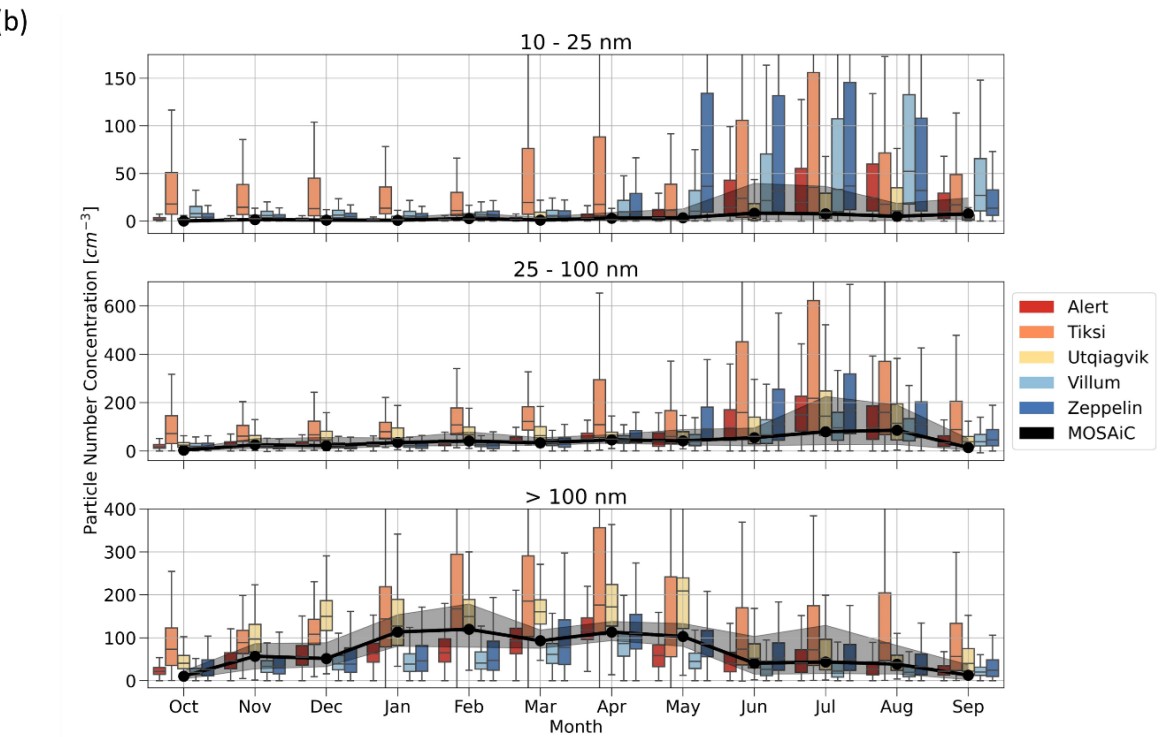





**Figure 9: Comparison of (a) PNSD and (b) PNC between MOSAiC and the long-term record of aerosol observations from land-based sites across the Arctic.** (a) The PNSD data for all sites is presented as monthly median values, and the shaded region indicates the interquartile range of the PNSD data from the MOSAiC campaign. (b) The box plots show the median monthly PNC and interquartile range at each site, where the particle concentrations are separated into three size modes (particle diameters of $10 - 25$ nm, $25 - 100$ nm, and $> 100$ nm). The MOSAiC data is presented as the solid black line, where the shaded region represents the interquartile range. The data from the land-based sites spans several decades, where the timespan varies depending on the site (Alert: 2011–2018, Tiksi: 2010–2018, Utqiagvik: 2007–2015, Villum: 2010–2020, Zeppelin: 2010–2020).

In general, the annual cycle in the PNSD, PNC, and BC observations over the sea ice in the central Arctic during MOSAiC agrees well with the various land-based sites across the Arctic. One of the most notable differences between MOSAiC and the land-based sites is the timing of the peak in Arctic Haze. During MOSAiC, the accumulation mode PNC peaked during February, whereas the accumulation mode peaked in April for all land-based sites, except for Tiksi and Utqiagvik. Accumulation mode aerosols peak in March and May at Tiksi and Utqiagvik, respectively (Fig. 9a). In addition to the higher accumulation mode concentrations, the shape of the MOSAiC PNSD observations during January and February differs compared to the other sites, whereas the PNSDs and PNCs are remarkably similar in shape and magnitude between MOSAiC and the long-term aerosol averages at Alert, Villum, and Zeppelin during the rest of the Arctic Haze period (March – May). Furthermore, the BC mass concentrations for January and February during MOSAiC are higher than the long-term averages at Utqiagvik, Villum, Zeppelin, and Gruvebadet (Fig. 10). The higher BC mass concentrations at Tiksi and Kevo may again be explained according to the sites' proximity to emission sources (Schmale et al., 2022), as also identified by the prevailing BC source regions in the inverse model. The particularly high BC mass concentrations observed at Kevo can be attributed to the length of the timeseries, which started in the 1960 when BC mass concentrations were much higher (Dutkiewicz et al., 2014; Schmale et al., 2022). Given that the MOSAiC dataset is only available for a single year, the different behavior in the accumulation mode aerosol and BC mass concentrations in January and February may be due to the interannual variability of Arctic Haze. We also must consider the ship's location, as the ship drifted from a location that may receive less aged pollution from Eurasia (Central Arctic Ocean) to one where the transport from polluted source regions takes longer (thus more dilution, i.e., Fram Strait) during the Arctic Haze season (Stohl, 2006).





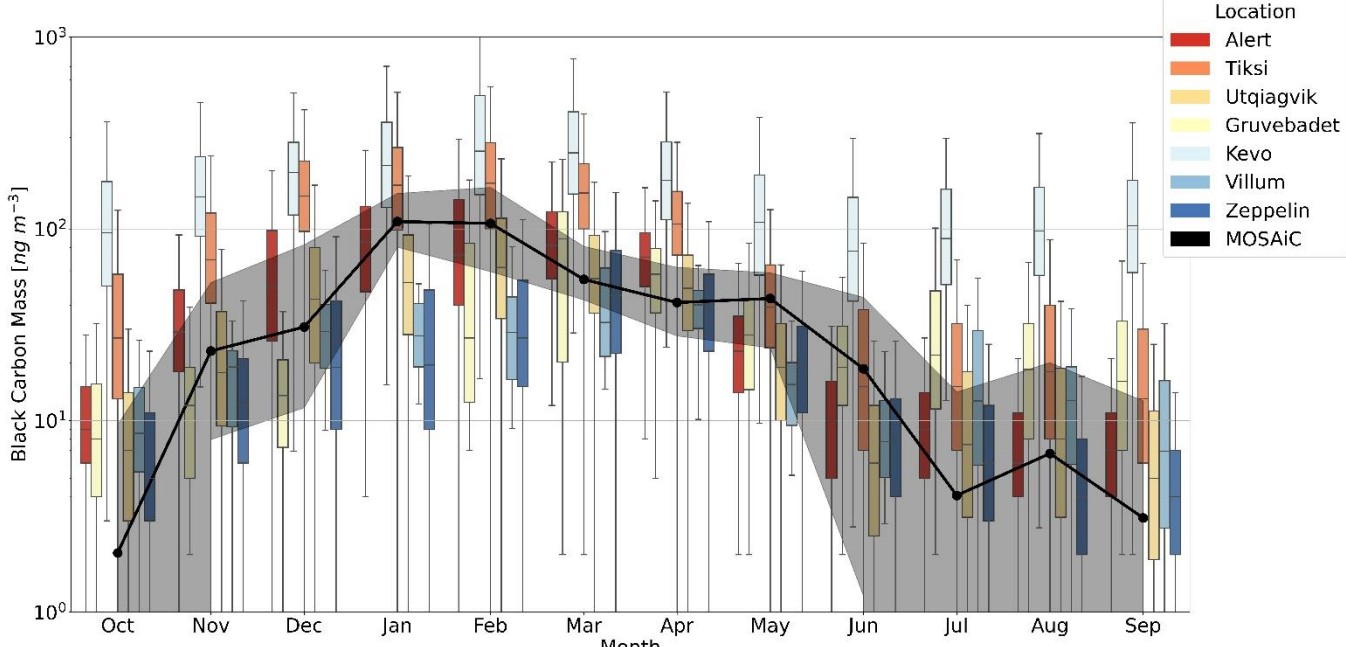

**Figure 10: A comparison of BC mass concentrations in the Arctic between land-based observatories and MOSAiC**. The MOSAiC
BC mass concentrations (black line, Aethalometer) are displayed as monthly medians, and the shaded region shows the interquartile range.
The boxes for the various sites represent BC and EC mass concentrations spanning several decades, where the time range varies by site
(Alert: 1989–2017, Gruvebadet: 2010–2018, Kevo: 1964–2010, Tiksi: 2009–2018, Utqiagvik: 1991–2019, Villum: 2009–2018, Zeppelin:
2001–2017). See Section 2.3 for details on the BC measurement methods, and refer the Schmale et al. (2022) for details on the long term
measurements from the land-based sites.

### 3.5.2 Comparison during the MOSAiC year

The comparison between the MOSAiC observations and the land-based sites during the year of the MOSAiC campaign (2019
– 2020) was used to further investigate and evaluate interannual variability (Fig. 11). At the time of this study, quality-
controlled PNSD datasets from the land-based sites during this period were only available from Alert, Villum, and Zeppelin.
The comparison shows that the shape and magnitude of the PNSDs during the Arctic Haze season are much more similar than
the comparison between MOSAiC and the climatological aerosol data in Fig. 9a, especially for January and February. The
PNC comparison during the MOSAiC year in Fig. 11b also identifies that accumulation mode aerosol reached an annual
maximum in February at Zeppelin, which is consistent with the MOSAiC observations. While the accumulation mode PNC
does not achieve an annual maximum during February at Alert and Villum, there are indeed enhancements in the accumulation
mode PNC at these sites during February 2020 compared to the long-term observations (Fig. 12). These results demonstrate
that the differences in Arctic Haze between the MOSAiC measurements and the climatological aerosol data from the land-
based sites are primarily due to interannual variability.





(a)

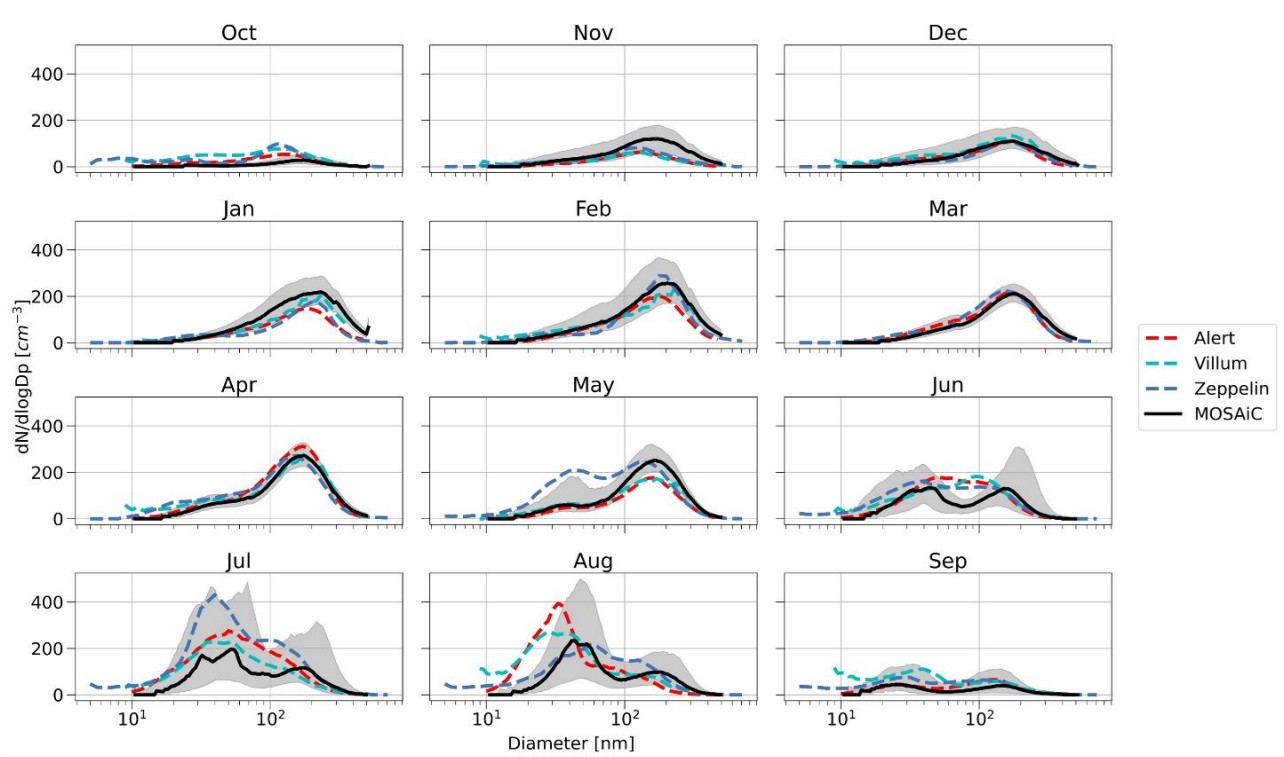

(b)

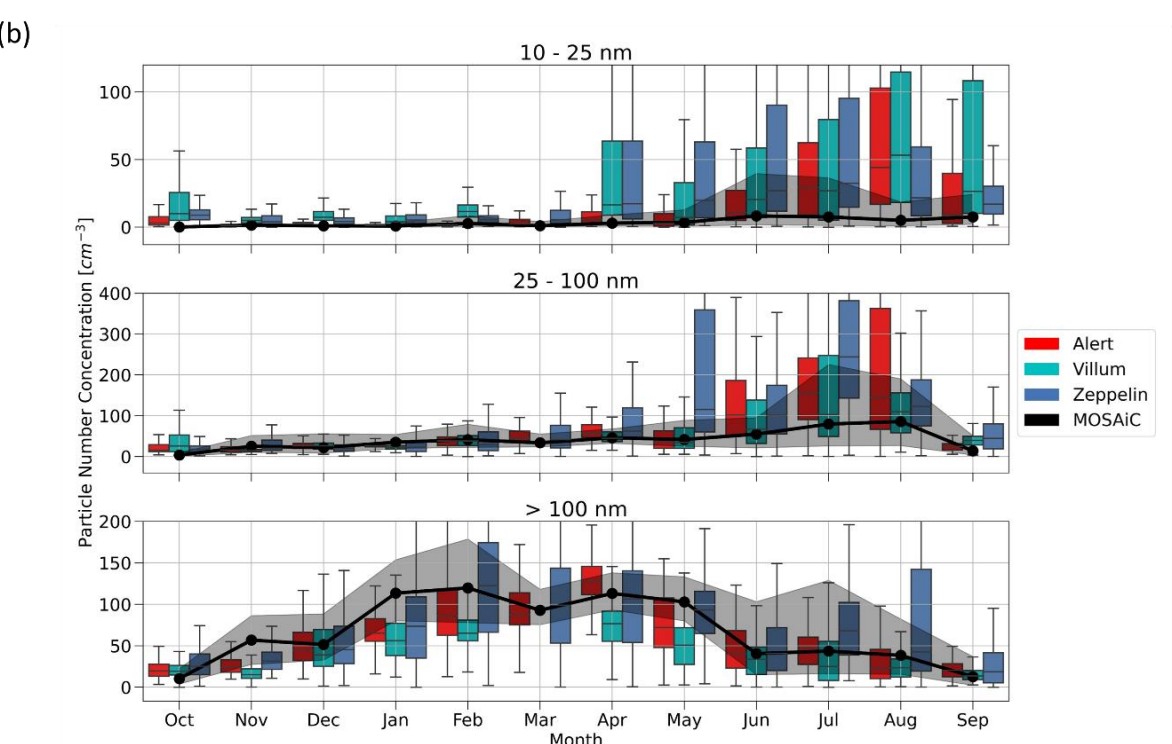





**Figure 11: Comparison of (a) PNSD and (b) PNC between MOSAiC and land-based sites during the year of the MOSAiC campaign.**
(a) the PNSD data is presented as monthly medians, and the shaded regions show the interquartile range of the MOSAiC data. (b) The box plots show the monthly median PNC data and interquartile range from the land-based sites for the three size modes (particle diameters of 10 – 25 nm, 25 – 100 nm, and > 100 nm). The MOSAiC data is presented as the solid black line, where the shaded region shows the interquartile range.

We can attribute much of the interannual variability in Arctic Haze during the MOSAiC year to the Arctic Oscillation (AO). The AO is a wintertime phenomenon that has two different phases, called the negative and positive phases. The negative phase is characterized by high-pressure anomalies in the central Arctic region, which facilitates air mass transport from the central Arctic to regions further south. In contrast, the positive phase exhibits low-pressure anomalies in the polar cap surrounded by a ring of high pressure in the midlatitudes, leading to potentially more efficient transport of air masses from lower latitudes

into the Arctic. A record-breaking positive phase of the AO was observed during January – March 2020 (Lawrence et al., 2020). This record-breaking positive phase in the AO corresponds with the earlier timing and higher intensity of Arctic Haze during MOSAiC, as observed in the peak accumulation mode PNC and BC mass concentrations during January and February 2020. To further investigate the influence of the record-breaking AO on observations across the Arctic, Fig. 12 shows a comparison between the averages from the MOSAiC year and the past decade for the AO, accumulation mode PNCs, and EC

mass concentrations at Villum. In general, the concentrations from January – March are higher than the average over the past decade, which shows that the effect of the record-breaking AO was influential on the aerosol across the Arctic region. The effect of the AO is less pronounced at Alert presumably due to dilution, however, we still observed an increase in the median accumulation mode concentration in February (and the upper quartile range) compared to the previous 10 years, suggesting that the AO was indeed influential at Alert. Further support for the influence of the AO across the Arctic is demonstrated by

BC observations during winter 2020 at Gruvebadet and Zeppelin, which show peak BC mass concentrations during January and February (Fig. 3a in Pasquier et al., 2022). These results agree with previous studies that have also observed that the positive phase of the AO can yield elevated pollution transport into the Arctic during this time of year (Eckhardt et al., 2003; Duncan and Bey, 2004; Di Pierro et al., 2013). In addition, previous research suggests that the positive phase of the AO may increase in frequency in a warming climate (Shindell et al., 1999), potentially further enhancing pollution transport into the

Arctic atmosphere in the future, which could counterbalance anticipated emission reductions.





**Figure 12: The influence of AO on aerosol observations during MOSAiC.** The five panels show the year of the MOSAiC campaign (2019 – 2020) compared to the trend over the past decade (2010 – 2019) for (a) the PNC > 100 nm at Alert, (b) the PNC > 100 nm at Villum, (c) the BC mass concentration at Villum, (d) the PNC > 100 nm at Zeppelin, and (e) the AO index. The data is presented as the monthly medians, and the shaded regions show the interquartile range. It is important to note that AO is a wintertime phenomenon; the complete year is included in the plots for context.

### 3.5.3 Context for MOSAiC aerosol measurements

Given the large monetary and environmental costs associated with performing measurements over the central Arctic Ocean for a full year, it would be desirable to use the measurements from one of the long-term land-based sites to represent the aerosol over this region. Overall, the agreement between the PNSD data collected during MOSAiC and the land-based sites suggests that, at least for the current state of the Arctic, some of the land-based measurement sites may be able to generally estimate the aerosol over the larger central Arctic Ocean region. However, one of the main conclusions from Freud et al. (2017) is that one site is not able to represent the aerosol population over the Arctic as a whole, and therefore, this should be done with caution. Freud et al. (2017) observed site-specific differences that make each site distinct beyond inter-annual variability, and these

differences are based on the location of a site and particularly the proximity of a site to the Arctic Ocean. Furthermore, a recent study provides evidence that episodic events, such as warm air mass intrusions, may not be fully captured by land-based observatories (Dada et al., 2022), and which further highlights potential issues with extrapolating land-based data to the central Arctic Ocean.

Indeed, we also observed distinct differences in the PNSD data collected from within the sea-ice over the central Arctic Ocean during MOSAiC compared to the land-based sites. The most notable difference is the relative magnitude of particles in the different size modes during different seasons. In Figs. 9 & 11, we show that the summertime PNC in the nucleation and Aitken mode observations from the central Arctic Ocean is lower than both the long-term averages and the data from the MOSAiC year, while the concentration of accumulation mode particles trends higher in the winter. The same pattern is observed in the

BC mass concentrations presented in Fig 10. Since we are limited to a single year and the MOSAiC observations were collected on a mobile platform, we are unable to provide robust conclusions on the role of interannual variability and geographic location in these observations. However, these results present remarkable evidence that aerosol observations over the sea ice in the central Arctic are distinct from the land-based sites throughout the year. Therefore, due to the rapid rate of change in the Arctic environment and the trend of sea ice decline, it is possible that the processes controlling the PNSD may change due to changes

in meteorology and transport pathways (e.g., Heslin Rees et al., 2020, Pernov et al., 2022), and such changes may be most pronounced over the central Arctic Ocean, where secondary aerosol processes may become more prevalent due to enhanced biological activity (Levasseur, 2013).

## 4 Conclusions

In this study, we present the aerosol measurements from the central Arctic during the MOSAiC expedition (October 2019 –

September 2020) and compare these observations to land-based sites across the Arctic. These aerosol measurements represent the first continuous PNSD data collected over the sea ice in the central Arctic Ocean for an entire year, especially at such northern latitudes during winter. The analysis focuses on the annual cycle in the PNSD and uses FLEXPART simulations, inverse modeling, aerosol observations from land-based sites, and other environmental variables to provide context to the in-situ observations of aerosols over the sea ice.


In the central Arctic, the aerosol population that we observe throughout the year is a result of transport and secondary aerosol formation processes. It is important to note the influence of both anthropogenic and natural sources of aerosols over the central Arctic Ocean during the year. In general, the particles >100 nm in diameter originate from anthropogenic sources, where the secondary processes contribute to smaller particles. Overall, the annual cycle in the PNSD data presented here agrees with

previous studies of the seasonality of Arctic aerosol from land-based sites (Tunved et al., 2013; Freud et al., 2017; Croft et al., 2016; Pernov et al., 2022; Dall'Osto et al., 2018; Lange et al., 2018). In winter, we observed elevated accumulation mode





aerosol, or Arctic Haze, dominated by anthropogenic sources in Russia/Siberia, namely the Norilsk smelter region. The wintertime Arctic Haze signal in the PNSD is consistent with the seasonality of BC mass concentrations. In addition, we identify that an unusually early peak of Arctic haze (i.e., in January and February as opposed to March and April) can be
attributed to a record-breaking positive phase of the AO, and the results show that the AO influences the timing of particulate pollution transport across the entire Arctic region. In summer, nucleation and Aitken mode aerosol are most prevalent in the PNSD. Although we do not explicitly explore the specifics further in this work, we conclude that the higher PNC of smaller particles results from enhanced secondary aerosol formation processes due to local gas-phase emissions from the ocean and ice combined with active photochemistry, as previously described in the literature (e.g., Willis et al., 2018 and references
therein). The role of precursor gases and NPF processes during the MOSAiC campaign will be evaluated in separate studies.

Our data provide further observations from within the central Arctic over the sea ice, and for the current state of the Arctic environment, the land-based sites may be useful for estimating the aerosol population over the sea ice in the central Arctic. However, as different locations in the Arctic are sensitive to specific source regions and thus can have differing aerosol
characteristics (Freud et al., 2017), this is likely to change as the Arctic environment continues to warm (Schmale et al., 2021). For the time being, the aerosol population in the central Arctic, as opposed to the land-based sites, is mainly dominated by long-range transport rather than local emissions or formation, except for the fall when locally new particle formation is a key process (Baccarini et al., 2020). However, the summertime sea ice extent is reaching record minimums, and the fraction of multi-year ice is declining, meaning that more local emissions, particularly from microbial activity, can become more
important. Changes in these processes due to changes in the Arctic environment could lead to additional changes in Arctic aerosol, hence changes in cloud properties. In this context, aerosols can exert an important climate-forcing effect in the Arctic atmosphere. Clouds, which interact with radiation, can influence the climate system because they affect the surface energy budget. This influence is especially relevant for the Arctic climate considering Arctic Amplification, which is driven by sea ice-albedo feedback processes (Serreze and Barry, 2011; AMAP, 2021).


At the same time, anthropogenic activities are increasing in the Arctic (Ferrero et al., 2016; Schmale et al., 2018). Our results demonstrate that both anthropogenic activities and natural processes are influential on aerosol properties in the central Arctic throughout the year, and both processes are subject to change in a warmer Arctic. A process-level understanding of how these changes impact aerosol properties allows models to evaluate and predict interactions between climatic change, socio-economic
change, and aerosols' impact on radiative forcing. Moreover, this study provides a snapshot of the current central Arctic environment with respect to aerosols, which is critical during this time of rapid environmental changes in the Arctic. There are few in-situ observations of aerosols in the Central Arctic. Therefore, the observations presented in this work can help serve the modeling community to understand current and future Arctic climate scenarios and their impacts on Earth's larger climate system.





**Data availability**

All datasets used in this work that were obtained during the MOSAiC campaign will be made publicly available by January 1, 2023 via Pangaea (https://www.pangaea.de/) or are already publicly available on the Department of Energy Atmospheric Radiation Measurement program (ARM) user facility data discovery tool (https://adc.arm.gov/discovery/#/).

Data from the Pangaea archive includes:

- Meteorological observations from *Polarstern*:
  - Schmithüsen, Holger (2021): Continuous meteorological surface measurement during POLARSTERN cruise PS122/1. Alfred Wegener Institute, Helmholtz Centre for Polar and Marine Research, Bremerhaven, PANGAEA, https://doi.org/10.1594/PANGAEA.935221

  - Schmithüsen, Holger (2021): Continuous meteorological surface measurement during POLARSTERN cruise PS122/2. Alfred Wegener Institute, Helmholtz Centre for Polar and Marine Research, Bremerhaven, PANGAEA, https://doi.org/10.1594/PANGAEA.935222

  - Schmithüsen, Holger (2021): Continuous meteorological surface measurement during POLARSTERN cruise PS122/3. Alfred Wegener Institute, Helmholtz Centre for Polar and Marine Research, Bremerhaven, PANGAEA, https://doi.org/10.1594/PANGAEA.935223

  - Schmithüsen, Holger (2021): Continuous meteorological surface measurement during POLARSTERN cruise PS122/4. Alfred Wegener Institute, Helmholtz Centre for Polar and Marine Research, Bremerhaven, PANGAEA, https://doi.org/10.1594/PANGAEA.935224

  - Schmithüsen, Holger (2021): Continuous meteorological surface measurement during POLARSTERN cruise PS122/5. Alfred Wegener Institute, Helmholtz Centre for Polar and Marine Research, Bremerhaven, PANGAEA, https://doi.org/10.1594/PANGAEA.935225

- Black carbon (BC): DOI coming soon.
- Particle number concentration (CPC3025):
  - Beck, Ivo; Quéléver, Lauriane; Laurila, Tiia; Jokinen, Tuija; Schmale, Julia (2022): Continuous corrected particle number concentration data in 10 sec resolution, measured in the Swiss aerosol container during MOSAiC 2019/2020. PANGAEA, https://doi.org/10.1594/PANGAEA.941886

The ARM data includes:

- Kuang, C., Singh, A., and Howie, J.: Scanning mobility particle sizer (AOSSMPS), https://doi.org/10.5439/1476898.

- Kuang, C., Salwen, C., Boyer, M., and Singh, A.: Condensation Particle Counter (AOSCPCF), https://doi.org/10.5439/1046184.

The landbased PNSD data:





- Alert: Personal communication from Tak Chan and Sangeeta Sharma, 2022. For details, see Croft et al. (2016).

- Villum: Personal communication from Jakob Boyd Pernov, 2022. For details, see Nguyen et al. (2016).

- Zeppelin: Personal communication from Peter Tunved, 2022. For details, see Tunved et al. (2013).

- Tiksi: Personal communication from Eija Asmi, 2022. For details, see Asmi et al. (2016).

- Utqiagvik: Freud, Eyal; Krejci, Radovan; Tunved, Peter; Leaitch, W Richard; Nguyen, Quynh T; Massling, Andreas; Skov, Henrik; Barrie, Leonard (2017): Hourly mean homogenised (dry diameter range 20 to 500 nm) observations of aerosol number size distributions from station Barrow, 2007-09-20 to 2015-07-09. PANGAEA, https://doi.org/10.1594/PANGAEA.877329

The land-based BC data:

- Villum: Personal communications from Daniel Thomas and Andreas Massling, 2022. For details, see Thomas et al. (2022).
- All other BC datasets used in this work will be made publicly available on EBAS (http://ebas-data.nilu.no/). Refer to Schmale et al., 2022 for additional details.


The Arctic Oscillation (AO) data is publicly available from the NOAA and the National Weather Service: https://www.cpc.ncep.noaa.gov/products/precip/CWlink/daily_ao_index/ao.shtml.

An archive of the FLEXPART model output and quick looks for the whole campaign can be found at
https://img.univie.ac.at/webdata/mosaic.

**Author contributions**

MB wrote the manuscript.

MB led the analysis, with conceptual contributions from JS, TJ, DA, JBP, and HA.

LQ, IB, JS, TJ, TL, MB, and ZB conducted field measurements reported in this work.

DA performed inversion modeling and source region identification work.

JS, JBP compiled datasets from land-based sites across the arctic and contributed to their analysis.

HA, LD, IB, JS, and BH provided and assisted with the interpretation of various datasets.

MDO and DB provided the PNSD cluster analysis results.

SB, MD, and AS performed the FLEXPART simulations and assisted with their interpretation.

EA, AM, DCT, TC, SS, PT, RK, and HC provided quality-controlled data from the land-based sites.

MK, TP, JS, and MS provided funding for the campaign and data analysis.

All authors provided comments/revisions on the manuscript.

**Competing interests**

The authors declare that they have no conflict of interest.



**Acknowledgments**

We acknowledge funding from the Swiss National Sciences Foundation grant no. 188478 and the Swiss Polar Institute, the US DOE grant No. DE-SC0022046, European Union's Horizon 2020 research and innovation program under grant agreement No. 856612 and the Cyprus Government. JS holds the Ingvar Kamprad Chair, sponsored by Ferring Pharmaceuticals. Part of this project was funded by ERC grant (GASPARCON) and the Academy of Finland Flagship funding (grant No. 337552). The

authors thank the Laboratory of Atmospheric Chemistry at the Paul Scherrer Institute for their support, and we thank Janne Lampilahti, Markus Lampimäki, Tommy Chan, and the INAR technical staff for their assistance with measurement support during the campaign. We also extend a special thank you to all personnel who made the expedition possible through the operation of the R/V *Polarstern* during MOSAiC in 2019–2020 (AWI_PS122_00) (Nixdorf et al., 2021). For Utqiaġvik and Tiksi the authors would like to thank the operators for the care and feeding of chemical and optical instruments, the Arctic and

Antarctic Research Institute (AARI) for provision of Tiksi data, and Derek Hageman for his continued excellence in data acquisition, processing, and archiving of data. Measurements and data evaluation for aerosol optical properties at Utqiaġvik are supported by Department of Energy/Atmospheric Radiation Measurement User (ANL award no. 0F-60239), and Aethalometer measurements at Tiksi occurred through NOAA cooperation with Roshydromet that ended in 2018. We thank the CNR Gruvebadet team around Stefania Gilardoni for sharing the PSAP data. All other data used in this manuscript were

produced, as part of the international Multidisciplinary drifting Observatory for the Study of Arctic Climate (MOSAiC) with the tag MOSAiC20192020. A subset of data was obtained from the Atmospheric Radiation Measurement (ARM) User Facility, a U.S. Department of Energy (DOE) Office of Science User Facility Managed by the Biological and Environmental Research Program.

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
