# Peer review of "A full year of aerosol size distribution data from the central Arctic under an extreme positive Arctic Oscillation: Insights from the MOSAiC expedition"

_Atmospheric Chemistry and Physics, 2022_

## Author Comment (AC1)

**Authors' response to reviewers**

Reviewer comments in black, Author responses in red

**Reviewer #1**

This paper reports the aerosol size distribution and BC concentration measured during the unique MOSAIC experiment in the central Arctic. Most part of the paper is devoted to the size distribution. Most of the findings confirm the knowledge obtained up to now from the land stations distributed around the Arctic, but that interannual variability could bring to results appreciably different from year to year (and that the MOSAIC year has been quite peculiar).

The article is well written and I have only a couple of comments, as well as few minor annotations.

Thank you for the general positive feedback and other comments on the manuscript, which has greatly improved due to the reviewer comments and suggestions.

**1.1** The authors use the stability of the atmosphere as an argument to justify the advection and the deposition of aerosols, but they don't show any measurements of this parameter taken during the cruise. It would be possible to provide some evaluation for it?

Thank you for your comment. The reviewer is correct that we discuss the stability of the atmosphere to describe the differences in aerosol transport between the winter and summer months, where the stability plays a key role in depositional processes of aerosol. Recent work by Josef et al. (2022) describes the local atmospheric structure during the MOSAiC expedition, however, we do not show any measurements related to stability or wet scavenging in this work because the local stability and precipitation (i.e., as measured on the *Polarstern*) are not particularly informative in the overall aerosol transport. Rather, the stability of the air masses and precipitation during transport would provide the most utility in evaluating these processes on our aerosol observations. As the seasonal change in aerosol transport and the associated role of stability and precipitation are well known and well characterized (e.g., Barrie, 1986; Bradley et al., 1992; Stohl, 2006; Freud et al., 2017; Bozem et al., 2019), we did not explore these processes further and simply refer to the literature.

**1.2** When comparing MOSAIC measurements with long-term observation taken at land stations, PNC from Tiksi, Villum and Zeppelin look very different (higher) than those measured onboard Polarster, in particular for particles with d < 100 nm. In my opinion this is not sufficiently highlighted in the text. Same for Figure 11 for the 1 year comparison.

The reviewer is correct that the lower concentrations of sub-100 nm particles over the central Arctic gives an interesting insight that is observable in our comparison of the MOSAiC data to the land-based sites.

We mentioned this in the abstract on Line 45 – 46:

"In summer, the PNC of nucleation and Aitken mode aerosol is enhanced, but concentrations were notably lower in the central Arctic over the ice pack than at land-based sites further south.",

as well as in Section 3.5.3, Lines 613 – 620:

"The most notable difference is the relative magnitude of particles in the different size modes during different seasons. In Figs. 9 & 11, we show that the summertime PNC in the nucleation and Aitken mode observations from the central Arctic Ocean is lower than both the long-term averages and the data from the MOSAiC year, while the concentration of accumulation mode particles trends higher in the winter. The same pattern is observed in the BC mass concentrations presented in Fig 10. Since we are limited to a single year and the MOSAiC observations were collected on a mobile platform, we are unable to provide robust conclusions on the role of interannual variability and geographic location in these observations. However, these results present remarkable evidence that aerosol observations over the sea ice in the central Arctic are distinct from the land-based sites throughout the year."

However, you are correct that this is not strongly highlighted in sections 3.5.1 and 3.5.2. We have updated the text in these sections to make the point clearer in the discussion:

- Line 526 – 530 in section 3.5.1: "In general, the annual cycle in the PNSD, PNC, and BC observations over the sea ice in the central Arctic during MOSAiC agrees well with the various land-based sites across the Arctic, but there are key differences as well. One interesting observation from our comparison is the lower concentrations of particles < 100 nm during summer in the central Arctic, as discussed further in section 3.5.3. Another notable difference between MOSAiC and the land-based sites is the timing of the peak in Arctic Haze."
- Line 557 - 560 in section 3.5.2: "As noted in the comparison between MOSAiC and the long-term measurements, we also observed that the land-based sites in general have higher concentrations of sub-100 nm particles than the central Arctic in summer during the MOSAiC year, which is discussed further in section 3.5.3. The comparison also shows that…"

Specific comments:

**1.3** L197 I would suggest to move the link to PSAP Gruvebadet data in the "Data availability" section, together with all the others.

We agree, it makes more sense to include this link in the "Data availability" section.

Line 195 – 200 now reads as follows:

"These land-based sites and the corresponding time ranges include Alert, Nunavut, Canada (1989–2017, Aethalometer); the Gruvebadet Atmosphere Laboratory (Gruvebadet), Svalbard (2010–2018, Particle Soot Absorption Photometer (PSAP)); the Kevo Subarctic Research Station (Kevo), Finland (1964–2010, elemental carbon, thermal optical method); Tiksi, Russia (2009–2018, Aethalometer); Utqiaġvik, Alaska, United States (1991–2019, PSAP, Continuous Light Absorption Photometer (CLAP), Aethalometer); Villum, Greenland (2009–2018, elemental carbon, thermal optical method); and Zeppelin, Svalbard, Norway (2001–2017, aethalometer).

We added the following to the "Data availability" section:

"• Gruvebadet: The PSAP data is accessible on the Italian Arctic Data Center operated by the National Research Council of Italy: https://data.iadc.cnr.it/erddap/tabledap/ebc_2010_2020.html."

**1.4** L205-209 You cite Beck et al. 2022 three times in few rows. Maybe you can re-phrase in order to do it once or twice.

We have removed the citation of Beck et al. (2022) from the first sentence in this paragraph, which is a general statement.

Line 205 – 206 now reads as:

"One of the challenges associated with ship-based campaigns is contamination of the ambient sample by emissions from the ship stack or other local pollution sources."

**1.5** L295 Desxription of panel (b) is missing

The caption for Fig. 2 on Line 295 was changed to the following:

"**Figure 2: The seasonality of surface influence from air mass source regions using FLEXPART.** A geographical mask (a) was applied to the FLEXPART air tracer data to quantify the FES associated with each geographical source region (b).  The FES was determined from the FLEXPART air tracer data within the lowest 100 meters of the atmosphere and was used to identify the influence of different source regions on the observed aerosol throughout the year."

**1.6** L305 I don't see the meaning of showing the hourly averages in S4, considering the fact that they are done over 1 year period and in 1 year you have so different environmental conditions. This is partly acknowledge by the authors in the text.

We agree that the hourly averages are typically used to track diurnal trends from disaggregated daily averaged clusters, as described in Beddows et al. (2014), and that the diurnal trends are not meaningful in the context of MOSAiC. However, we have included the disaggregated hourly averages of the clusters in Fig. S4 as another way to show the variability in the PNSD in the clusters that is not otherwise visible from the daily average PNSD clusters show in Fig. 3.

**1.7** Figure 10 It is difficult to distinguish the different colors (e.g. Utqiagvik from Gruvabadet). In which order are reported the different stations in the figure? In my opinion is not the same as in the legend.

That is correct, the order of the station data in the figure does not match the order in the legend. We corrected this error by changing the order of the data from the stations in the figure to match the order in the legend, which improves the clarity of the data presentation. To further improve the readability of the figure, we also adjusted the saturation of the stations represented by yellow colors.

Here is the updated Fig. 10:

[Figure]

**Figure 1: A comparison of BC mass concentrations in the Arctic between land-based observatories and MOSAiC.** The MOSAiC BC mass concentrations (black line, Aethalometer) are displayed as monthly medians, and the shaded region shows the interquartile range. The boxes for the various sites represent BC and EC mass concentrations spanning several decades, where the time range varies by site (Alert: 1989–2017, Gruvebadet: 2010–2018, Kevo: 1964–2010, Tiksi: 2009–2018, Utqiagvik: 1991–2019, Villum: 2009–2018, Zeppelin: 2001–2017). See Section 2.3 for details on the BC measurement methods, and refer the Schmale et al. (2022) for details on the long-term measurements from the land-based sites.

References

Barrie, L. A.: Arctic air pollution: An overview of current knowledge, 20, 643–663, https://doi.org/https://doi.org/10.1016/0004-6981(86)90180-0, 1986.

Beck, I., Angot, H., Baccarini, A., Dada, L., Quéléver, L., Jokinen, T., Laurila, T., Lampimäki, M., Bukowiecki, N., Boyer, M., Gong, X., Gysel-Beer, M., Petäjä, T., Wang, J., and Schmale, J.: Automated identification of local contamination in remote atmospheric composition time series, Atmos. Meas. Tech., 15, 4195–4224, https://doi.org/10.5194/amt-15-4195-2022, 2022.

Beddows, D. C. S., Dall'Osto, M., Harrison, R. M., Kulmala, M., Asmi, A., Wiedensohler, A., Laj, P., Fjaeraa, A. M., Sellegri, K., Birmili, W., Bukowiecki, N., Weingartner, E., Baltensperger, U., Zdimal, V., Zikova, N., Putaud, J.-P., Marinoni, A., Tunved, P., Hansson, H.-C., Fiebig, M., Kivekäs, N., Swietlicki, E., Lihavainen, H., Asmi, E., Ulevicius, V., Aalto, P. P., Mihalopoulos, N., Kalivitis, N., Kalapov, I., Kiss, G., de Leeuw, G., Henzing, B., O'Dowd, C., Jennings, S. G., Flentje, H., Meinhardt, F., Ries, L., van der Gon, H. A. C., and Visschedijk, A. J. H.: Variations in tropospheric submicron particle size distributions across the European continent 2008–2009, 14, 4327–4348, https://doi.org/10.5194/acp-14-4327-2014, 2014.

Bozem, H., Hoor, P., Kunkel, D., Köllner, F., Schneider, J., Herber, A., Schulz, H., Leaitch, W. R., Aliabadi, A. A., Willis, M. D., Burkart, J., and Abbatt, J. P. D.: Characterization of transport regimes and the polar dome during Arctic spring and summer using in situ aircraft measurements, 19, 15049–15071, https://doi.org/10.5194/acp-19-15049-2019, 2019.

Bradley, R. S., Keimig, F. T., and Diaz, H. F.: Climatology of surface-based inversions in the North American Arctic, J. Geophys. Res., 97, 15699, https://doi.org/10.1029/92JD01451, 1992.

Freud, E., Krejci, R., Tunved, P., Leaitch, R., Nguyen, Q. T., Massling, A., Skov, H., and Barrie, L.: Pan-Arctic aerosol number size distributions: seasonality and transport patterns, 17, 8101–8128, https://doi.org/10.5194/acp-17-8101-2017, 2017.

Jozef, G., Cassano, J., Dahlke, S., and de Boer, G.: Testing the efficacy of atmospheric boundary layer height detection algorithms using uncrewed aircraft system data from MOSAiC, Atmos. Meas. Tech., 15, 4001–4022, https://doi.org/10.5194/amt-15-4001-2022, 2022.

Stohl, A.: Characteristics of atmospheric transport into the Arctic troposphere, 111, https://doi.org/https://doi.org/10.1029/2005JD006888, 2006.

**Reviewer #2**

**Summarisation**

This paper describes particle number size distribution (PNSD) and 'black carbon' observations from the Mosaic field study. They investigate seasonality and sources of the observed aerosol and relate the observations to other sites in the Arctic. They find that meteorological and dynamical conditions (temperature and the Arctic oscillation) have an important impact on the aerosol characteristics.

**General comments:**

**2.1** This manuscript does an excellent job of citing recent papers, but to my mind misses some of the initial work on which the more recent papers rely. For example, Quinn et al, JGR, 2002 discusses seasonal changes in aerosol sources at Utqiaġvik/Barrow, while Schmeisser et al, ACP, 2018 describes seasonal aerosol variability at 6 Arctic land-based sites.

Thank you for pointing out other relevant articles that are not cited in our work. The reviewer is correct, the two references that you suggested provide value in the context of this work, and we have added them to text (and references) as follows:

- Line 74 – 79: "Arctic Haze has been observed to occur during winter and persist through spring at several land-based sites across the Arctic (Heidam et al., 1999; Quinn et al., 2002; Quinn et al., 2007; Tunved et al., 2013; Freud et al., 2017; Sharma et al., 2019). Previous observations of aerosol size distributions around the Arctic show that Arctic Haze is dominated by accumulation mode aerosol, or particles > 100 nm in diameter, while summer features a dominant Aitken mode and a Hoppel minimum, which indicates the strong effect of cloud processing on the aerosol population (Quinn et al., 2002; Tunved et al., 2013; Croft et al., 2016; Asmi et al., 2016; Nguyen et al., 2016; Freud et al., 2017)."

- Line 610 – 613: "However, the main conclusions from Freud et al. (2017) and Schmeisser et al. (2018) are that one site is not able to represent the aerosol population over the Arctic as a whole, and therefore, this should be done with caution. Both studies observed site-specific differences that make each site distinct beyond inter-annual variability…"

**2.2** Lines 207-209 - Could your filtering be more restrictive than the filtering applied at ground sites? Perhaps if your filtering technique/algorithm had been applied to the land-based datasets they would not exhibit higher values than the Mosaic dataset. Also, i am not sure it's of use to provide the values for constants 'm' and 'a' without any further details. Just refer to Beck publication (unless the Beck publication is more general and doesn't use those specific values?)

This is a helpful comment. It is relevant to point out that the method of filtering pollution could influence the data, and the methods used among different studies can vary considerably. For this reason, we used the pollution detection algorithm developed by Beck et al. (2022) for the MOSAiC dataset, as it is a traceable and reproducible method for flagging pollution from a dataset. An important feature of the method in Beck et al. (2022) is that it provides user flexibility in various input parameters of the algorithm, and thus, specific values for 'm', 'a', and the median time are useful to reproduce the final, pollution-free SMPS dataset used in this analysis, which is why we chose to include this information in the methods.

Furthermore, Beck et al. (2022) presented a comparison of the pollution detection algorithm (used also in this work) compared to a visual inspection method to remove pollution from the particle concentration data. The comparison showed that the pollution detection algorithm agreed with the visual inspection method for 94% of the data during the year, which provides confidence in the performance of the pollution detection method used herein.

It is important to note that pollution was already removed from the land-based PNSD datasets that were provided to us in this work. These datasets have been quality controlled by the station PIs, who have detailed knowledge about local pollution sources and how to detect their influence on the PNSD. While there are differences in methodology between sites, these datasets have undergone cleaning which is specific to each site and represents (to the best of the data providers knowledge) pollution free, quality-controlled data.

Finally, we utilized monthly median particle number concentration values for the comparison between MOSAiC and the land-based sites, as monthly median values are more robust and less prone to bias from transient spikes than monthly averages or data at a higher time resolution. Therefore, it is unlikely that the different methods for data filtering are a major cause of the difference in the concentration levels between the ground sites and MOSAIC.

**2.3** Line 220-223 - how different were the results of masking from the two measurement containers? Were the inlets that different? I find that somewhat disturbing.

There are several aspects of the inlets that can influence the timing of pollution in the data. These aspects include location of the measurement container on the bow, inlet height, and the automatic purge blower installed on the ARM AOS inlet.

First, the two measurement containers were located on different sides of the bow, where differences in wind direction and turbulence around the superstructure of the ship could influence the measurements from the two containers differently (refer to Fig A2 in Beck et al., 2022 for a picture of the two containers and their inlets). It's also worth noting that the *Swiss* container was located on the starboard side of the ship that was closer to the MOSAiC central observatory, where logistic activities could have a more direct influence.

Concerning inlet height, the ARM AOS inlet was 5 meters in length, while the *Swiss* container inlet was considerably shorter (1.5 meters). For the same reasons that were identified due to the differences in location on the bow, the differences in inlet height between the two containers can affect the timing of pollution in the data.

Another key difference, and the most important reason for using separate pollution masks for the two containers, is the use of an automatic inlet purge blower system that was installed on the ARM AOS inlet. A discussion of the purge blower and its implications in the ARM AOS data is given in the Supplemental Information, which we will summarize here. In the event of pollution, the purge blower was able to provide a high volume of clean, particle free air directly into the AOS inlet stack to displace the pollution. The purge blower was intended to automatically trigger due to elevated CO mixing ratios due to contamination from the ship stack or other anthropogenic pollution sources, however, there was very little CO associated with the ship stack pollution from the *Polarstern* during MOSAiC (see Beck et al. (2022)). Therefore, the purge blower was turned on manually by operators on the ship during extended pollution episodes to protect the instrumentation from potential contamination associated with the high particle load from the ship stack, and the data are no longer

representative of ambient conditions when the purge blower was enabled and were removed. A similar mechanism was not in place on the inlet in the *Swiss* container, meaning that a separate pollution mask was necessary to remove polluted data from instruments located in the *Swiss* container during periods where the purge blower was enabled in the ARM AOS.

**2.4** Figure 2 - How would the results in Figure 2b differ if used a midpoint of the entire cruise rather than the cruise track? Same question for Figure 7.

We can speculate that there would be some small changes to the FES influence using a midpoint rather than the cruise track as the endpoint in the FLEXPART simulations, such as the example we gave about enhanced continental influence due to the ship's location in Lines 289 – 293. However, we would not expect the overall conclusions about seasonality in the aerosol PNSD to change, as our observations agree well with the seasonal trends from previous studies from land-based stations in the Artic, as discussed in Section 3.5.3. We also do not expect our conclusions on the role of Arctic Oscillation during the MOSAiC year would change by using the midpoint, as the influence can be noted at other land-based sites as well, as shown in Fig. 12.

**2.5** Figure 4 - What causes the abrupt shift in PNSD in June 2020? I would expect more of a gradient. Is it due to course changes - the excursion to Svalbard? or perhaps not enough representative days to get a monthly average?

It is important to note that Figure 4 represents the monthly median PNSD, which does have limitations in showing gradients that could occur in the PNSD data from mid-May to mid-June, for example. For this reason, we refer to the PNSD cluster analysis, to evaluate the seasonal transition observed to occur starting in May, as described in the paragraph starting on Line 390 in section 3.3.2.:

"May represents the time when the Arctic aerosol processes undergo a seasonal transition; the occurrence of the accumulation mode from Arctic Haze subsides, and the nucleation and clean PNSD clusters become more prevalent during the month (Fig. 5). As the Arctic environment continues to change, these temperature-dependent transport processes may influence the future seasonality and prevalence of anthropogenic pollution as well as the transition to natural aerosols in the central Arctic (Browse et al., 2012), which can yield important implications for radiative processes in the Arctic atmosphere (Shindell and Faluvegi, 2009)."

Concerning data coverage, Fig. S2 presents the percentage of data remaining during each month after applying the pollution mask, and it shows that 37 % of the data remains in May, which is comparable to the average for all months during the campaign (average = 42%, median = 38%, standard deviation = 17%). Figure S5 depicts the PNSD data as daily medians, which provides another perspective of the data coverage during the year, and the higher time resolution allows for temporal variability, such as the transition during the month of May, to be observed more clearly. Please note that while June does indeed have less data coverage compared the other months (23%), Fig. S5 also shows that this is mostly due to poor data coverage at the end of the June.

**2.6** The manuscript defines the Central Arctic as above 80 N. Utqiaġvik, Tiksi and Kevo, while in the Arctic, are all further south and closer to 70 N. Because of that, it seems not too surprising that they

are less representative of the 'Central Arctic' than the more northerly sites.  Their lower latitudes not only make them closer to continental source regions (Europe/N. America/N. Asia), but it also makes them relatively more temperate, resulting in more local/regional populations and sources and warmer temperatures.

Thank you for your comment and for providing your insights on this topic. We agree that the results are not too surprising, but this observation is only made unambiguously with a direct comparison using data from the central Arctic, as we did in our analysis. To further emphasize these points, we updated this discussion in lines 511 – 515 in the text as follows:

"The largest differences in the long-term observations among the land-based sites occur in Tiksi and Utqiaġvik, which typically have more variation in the shape of the PNSD and a higher PNC throughout the year. These differences are likely due to the proximity of these sites to large emission sources, as well as their more southerly locations in the Arctic compared to the other sites, which makes them relatively more temperate. As a result, the data from Tiksi and Utqiaġvik may not be as representative of the baseline aerosol concentrations across the larger Arctic region as the other sites…"

**2.7** The manuscript discusses how the size distribution changes during the Mosaic measurements as a function of air temperature (e.g. Fig 4).  Is air temperature seasonality also relevant to the observed seasonality of size distributions at the various ground-based sites mentioned in the manuscript?

Due to similar trends in the seasonal phenomenon of Arctic Haze at the arctic sites, we can speculate that air temperature does have an influence on the seasonality of the size distributions at the land-based sites as well, but we do not explore this directly in this work. As discussed in section 3.3.2, the key insight from the temperature data from MOSAiC was that the ambient air temperature corresponded well with the timing of Arctic Haze, due to the enhanced stability previously observed during the cold months during winter and spring that influences aerosol transport and lifetime. Since the stability has been shown to result from transport of air masses over cold, snow-covered regions and breaks down when warming occurs over continental regions in summer (Bradley et al., 1992; Klonecki et al., 2003), it is likely that this same effect would have an influence at the land-based sites as well. In a more general sense, warming in the Arctic has been shown to affect large scale atmospheric circulation patterns and meteorology (e.g., Lee et al., 2015; Francis et al., 2017), and such changes to meteorology and transport patterns have already been identified to influence aerosol processes at Villum (Pernov et al., 2022) and Zeppelin (Heslin Rees et al., 2020).

**Data source and attribution of the 'Utqiaġvik' data**

**2.8** The proper name of the location at which the Utqiaġvik data were acquired is the NOAA Barrow Atmospheric Baseline Observatory which is near the village of Utqiaġvik in Alaska: https://gml.noaa.gov/news/brw_dedication.html

It would be appropriate to mention in the text the station's full name at least once and to note that it is a NOAA site since neither the 'Utqiaġvik' BC or PNSD measurements would be possible without the infrastructure provided by NOAA - particularly since there are no NOAA co-authors on the

manuscript. Schmale et al 2022 referred to this data as 'Utqiaġvik /Barrow'. NOAA is not mentioned anywhere in the manuscript in relation to the measurements. Readers should not have to dig into datasets to find out exactly where the measurements were made.

This is indeed important. We acknowledge that we should have stated NOAA's connection with the data from Utqiaġvik and referred to the NOAA Barrow Atmospheric Baseline Observatory more clearly. Upon the first mention of the station in line 179, we now introduce the site as:

"…the NOAA Barrow Atmospheric Baseline Observatory, near the village of Utqiaġvik, Alaska (referred to hereafter as Utqiaġvik)…"

Concerning the BC data, permission was obtained from Betsy Andrews to use the same dataset presented in Schmale et al. (2022), including the acknowledgment in Line 752 – 758.

**2.9** Related - it's unclear to me if the other Arctic ground sites are represented by co-authors on the manuscript's author list - if not, then it would be good to ask someone from the unrepresented sites how they want their sites to be referred to.

The authors cite Freud et al (2017) on the PANGAEA archive as the source of the Utqiaġvik PNSD data. These data were generated by IfT in Leipzig (Ali Wiedensohler's group) from instruments deployed at NOAA's Barrow Atmospheric Observatory near Utqiaġvik. I do not see an IfT co-author cited for this dataset in the data availability section. Before using this dataset, I would recommend that the authors check with the data generators (IfT) on the quality and attribution of these data.

When bringing together data from the various stations, we relied first and foremost on already published datasets, openly accessible through data repositories. We also checked whether there were any indications submitted together with the data of how data originators wanted to be acknowledged or involved. Datasets that were not yet published are represented by co-authors.

Specifically for Utqiaġvik, as mentioned, the aerosol PNSD data that was used in this work was obtained from the PANGAEA archive and is cited in Line 715. There were several attempts to contact Eyal Freud directly, but there was no response. We also attempted to contact Kerri Pratt, the corresponding author of Kolesar et al. (2017), with no response. Previously, we did not reach out further to anyone from TROPOS since there is no mention of TROPOS in association with the dataset, and the dataset is publicly available and citable on PANGAEA.

Following the suggestion of the reviewer, we reached out to Alfred Wiedensohler and Kay Weinhold for further clarification on the data and its treatment and acknowledgement in this work. They were also unaware of the dataset published by Freud et al. (2017) on PANGAEA, but they did conclude that they were responsible for the measurements, as Freud et al. (2017) states the following in reference to the Utqiaġvik data: "…only recently a custom-built SMPS, measuring aerosol dry diameters in the range of 10 to 990 nm, has been installed there by the Leibniz Institute for Tropospheric Research (IfT) in Leipzig, Germany." They are now included as coauthors on the current work.

**2.10** The Freud paper which developed the data set on the the PANGAEA archive says they obtained the raw Utqiaġvik PNSD data and did some filtering based on wind direction. That sounds different than the data treatment described in Kolesar et al 2017 (written by scientists from Ift) which is cited by this manuscript to describe data treatment of the Utqiaġvik PNSD data.

Thank you for pointing out the differences in descriptions of the data treatment. The reviewer is correct that it is better to cite Freud et al. (2017) instead of Kolesar et al. (2017) in this context. The citation in Line 181 – 182 was changed accordingly.

**Minor comments**

**2.11** Line 51 "...surface energy budget and has..." -->"...surface energy budget and have..."

Corrected.

**2.12** Line 79-80 "At the same time this demonstrates the importance of aerosol particles in cloud formation processes." Unclear what this sentence refers to as demonstrating the importance of aerosol particles in cloud formation processes.

The reviewer is correct. This sentence, in its current state, is vague and does not add value to the discussion. It has been removed from Line 79 – 80, which is now as follows:

"In addition, previous research suggests that the anthropogenic pollution associated with Arctic Haze can enhance longwave radiation emission from Arctic clouds, leading to surface warming (Garrett and Zhao; 2006)."

**2.13** Line 100 - Might make sense to caveat the identification of anthropogenic and natural sources. Some sources (fires/dust) might be enhanced by human-caused warming, drying or land-management practices.

This is a good suggestion.

We have added "…in response to anthropogenic-induced warming…" to Line 97 and "…whereby the latter are indicators of climate change feedback processes from anthropogenic-induced warming…" to Line 100 & 101.

**2.14** Line 111 - what is meant by 'targeted monthly approach'

We have changed the text on Line 111 to clarify:

"We use monthly averages (medians) to characterize…."

**2.15** Line 121 - the central Arctic is mentioned in the first paragraph of the methods section but then not actually defined until lines 135-136. Move definition earlier.

The reviewer is correct. We have changed the text introducing the MOSAiC expedition in Lines 120 – 123 to refer to the Arctic Ocean more generally:

"The MOSAiC campaign was designed to address the scarcity of data available from the Arctic Ocean region. In fact, the MOSAiC expedition was the most comprehensive expedition in the Arctic Ocean

in history, and researchers evaluated the Arctic Ocean environment for an entire year from various perspectives, including oceanography, sea ice dynamics, biology, meteorology, and atmospheric physics and chemistry."

**2.16** Line 175- Kuang 2016b is not in references (neither is Kuang 2016a)

Thank you for identifying this error. Kuang 2016a & Kuang 2016b have been added to the references at Line 983:

"Kuang, C.: Condensation particle counter (CPC) instrument handbook. ARM Tech. Rep. DOE/SC-ARM-TR-145, 22 pp., https://doi.org/10.2172/1245983, 2016a.

Kuang, C.: TSI Model 3936 scanning mobility particle spectrometer instrument handbook. ARM Tech. Rep. DOE/SC-ARM-TR-147, 22 pp., https://doi.org/10.2172/1245993, 2016b."

**2.17** Line 179 - please use the proper diacritical marks for Utqiaġvik - there is a dot on top of the g.

All references to Utqiaġvik are updated to include the correct spelling with the diacritical marks, including in the figures.

**2.18** Line 192 - "...while this measurement represents equivalent BC (eBC)." --> "...while properly this measurement should be referred to as equivalent BC (eBC)."

Line 192 was rephrased according to your suggestion for improved clarity.

**2.19** Line 198-200 - The full time series of NOAA's Barrow Atmospheric Baseline Observatory absorption data from PSAP and CLAP for 1997-2021 (which is what the BC data in the Schmale et al data sets is calculated from) is available from both from NOAA: https://gml.noaa.gov/aftp/aerosol/brw/ and from EBAS: https://ebas.nilu.no.

This would enable you to include Utqiaġvik BC data in Figure 12.

The BC data from Utqiaġvik in Schmale et al. (2022) only includes data up to 2019, and at the time of writing, we only agreed to use the dataset that was published in Schmale et al. (2022) in this work.

Please refer to comment 2.30 for discussion of Figure 12.

**2.20** Line 254 - How many 5 min scans were required to have an acceptable hourly mean? How many acceptable hourly scans were required to have an acceptable daily mean?

Thank you for your comment. We were not clear in specifying that the cluster analysis was based on daily means of the 5-minute pollution-filtered data set. The hourly averages were only used for the disaggregation of the daily-averaged clusters, as shown in Fig. S4. Each daily mean can therefore contain up to 288 measurements, and it does not make sense to set exclusion criteria based on our observations that the clusters results were on average within 10% of each other when progressively excluding days with less than 10%, 20%, 30%, 40%, 50%, 60%, and 70% of the 5-minute

measurements. Furthermore, despite the ~10% error within a given cluster when setting exclusion criteria, a clear separation between the cluster centers (the mean PNSD plots) was maintained.

This has been clarified in the text:

- Line 255 – 264: "The PNSD measurements provided by the SMPS from MOSAiC were aggregated into daily (297) arithmetic means. The daily PNSD measurements were then normalized to their vector length, and a cluster analysis was performed using k-means according to the Hartigan-Wong method (Hartigan and Wong, 1979), which is an established method for evaluating aerosol size distribution characteristics (Beddows et al., 2009; Beddows et al., 2014; Freud et al., 2017, Pernov et al. 2022). By normalizing the data to the vector length, the shape of the PNSD, rather than its magnitude, was clustered. Since the cluster analysis was performed on the daily PNSD measurements, the resulting clusters show the typical PNSD for a day, where each day of PNSD data is assigned to a single cluster. Note that we did not set exclusion criteria for the amount of 5-minute data available on a given day when calculating the daily average PNSDs, as exclusion criteria only served to limit the amount of data without any notable changes to the cluster definitions. The cluster analysis was carried out using 2 to 10 clusters, of which the 9-cluster output best described the data."

**2.21** Line 255 - what is meant by 'vector length' in this context?

The vector length is the "length of the vector" or magnitude of the vector |x| (i.e., $\sqrt{x_1{}^2 + x_2{}^2 + \cdots + x_n{}^2}$ ). This has been previously described in the literature that we cite for the clustering analysis. For more details, refer to the supporting information of Beddows et al. (2009) (https://pubs.acs.org/doi/abs/10.1021/es803121t).

**2.22** Line 267 - what wavelength is the short-wave DW radiation?

The specifications from the Kipp & Zonen CM11 manual report a spectral range from 310 - 2800 nm.

**2.23** Line 279-280 - North America appears to be most important in May-August based on Figure 2b which contradicts the statement "Much lower contributions from the continental regions were observed during May - July."

While it is true that the contribution from North America does increase in May – July compared to the winter months, the intended meaning of the statement is that the combined influence from all continental regions (North America + North Asia + Europe) is less during May - July. To clarify the statement, we have rephrased Line 279-280 as follows:

"Most notably, the combined influence from these continental regions further south resulted in higher overall continental FES values from November - April, especially from North Asia. The combined influence from continental regions was observed to be much lower during May – July."

**2.24** Line 285 - "...a shrinking effect of the polar dome, or a meteorological..." --> "...shrinking of the polar dome, a meteorological..."

The sentence in Line 285 was rephrased according to your suggestion.

**2.25** Figure 2 - caption - describe the mask first or switch the position of figure a and b.

Thank you for your comment on the caption for Fig. 2. The caption has been reworded as follows:

"**Figure 2: The seasonality of surface influence from air mass source regions using FLEXPART**. A geographical mask (a) was applied to the FLEXPART air tracer data to quantify the FES associated with each geographical source region (b). The FES was determined from the FLEXPART air tracer data within the lowest 100 meters of the atmosphere and was used to identify the influence of different source regions on the observed aerosol throughout the year."

**2.26** Figure 3 - what is the variability in the average size distributions?  Show with shading

We added the interquartile range of the average size distributions for the cluster to Figure 3. The updated figure is now presented as follows:

[Figure]

"**Figure 2: The average distributions of the PNSD clusters.** The particle concentration in units of dN/dlogD$_p$ is presented for the average of all days in each cluster according to the particle size. The shaded regions show the interquartile range associated with the PNSD data for each cluster type. Note that the nucleation and accumulation mode clusters presented here are the result of manually grouping the two nucleation mode clusters and four accumulation mode clusters into a single nucleation and accumulation mode cluster, respectively. Refer to Fig. S3 for the original 9 cluster output."

**2.27** Line 375 - "Stohl., 2006" --> Stohl, 2006"

Corrected.

**2.28** Line 390 - "May itself represents..." --> "May represents..."

Corrected.

**2.29** Figure 8 - use same color scheme as Fig 2b (North America is a different color)

Thank you for identifying this error with the color scheme in Fig. 8. Figure 8 has been updated in the text:

[Figure]

**2.30** Figure 12 - Include Utqiaġvik BC - can get the absorption data for year of Mosaic from EBAS or from NOAA. Just need to apply whatever mass absorption efficiency Schmale et al 2022 applied to convert absorption into BC concentrations.

According to your suggestion, we have obtained eBC data from Utqiaġvik during the MOSAiC year and included it in the comparison in Fig. 12.

The new figure is as follows:

[Figure]

**Figure 3: The influence of AO on aerosol observations during MOSAiC.** The six panels show the year of the MOSAiC campaign (2019 – 2020) compared to the trend over the past decade (2010 – 2019) for (a) the PNC > 100 nm at Alert, (b) BC mass concentration at Utqiaġvik, (c) the PNC > 100 nm at Villum, (d) the BC mass concentration at Villum, (e) the PNC > 100 nm at Zeppelin, and (e) the AO index. The data is presented as the monthly medians, and the shaded regions show the interquartile range. It is important to note that AO is a wintertime phenomenon; the complete year is included in the plots for context.

This data was provided by Elisabeth Andrews, who is cited in the data availability section as such on Line 735:

> "• NOAA Barrow Atmospheric Baseline Observatory (Utqiaġvik) during 2020: Personal communication from Elisabeth Andrews, 2022. For processing details, see Schmale et al. (2022)."

Various sections of the text were updated to account for the addition of the eBC data from Utqiaġvik:

- Line 202 – 203: "…Utqiaġvik, Alaska, United States (1991–2020, PSAP, Continuous Light Absorption Photometer (CLAP), Aethalometer); …"

- Line 563 – 565: "At the time of this study, quality-controlled PNSD datasets from the land-based sites during this period were only available from Alert, Villum, and Zeppelin, and BC data was available from Utqiaġvik and Villum."

- Line 593 - 603: "To further investigate the influence of the record-breaking AO on observations across the Arctic, Fig. 12 shows a comparison between the averages from the MOSAiC year and the past decade for the AO, accumulation mode PNCs, and BC mass concentrations at Utqiaġvik and Villum. In general, the concentrations from January – March are higher than the average over the past decade, which shows that the effect of the record-breaking AO was influential on the aerosol across a large part of the Arctic region. There is one notable exception in this analysis: Utqiaġvik. As noted previously, Utqiaġvik, while still located in the Arctic, is further south than the other sites in Fig. 12, which makes it more temperate and potentially less representative of the aerosol within the central Arctic region. The more southern location of Utqiaġvik could also mean that it is not directly affected by enhanced pollution transport related to the AO, which is consistent with both the sea level pressure anomalies presented in Fig. 5a in Lawrence et al. (2020) and previous observations that pollution transport from Eurasia is favored during the Arctic Haze season (Stohl, 2006)."

References

Beck, I., Angot, H., Baccarini, A., Dada, L., Quéléver, L., Jokinen, T., Laurila, T., Lampimäki, M., Bukowiecki, N., Boyer, M., Gong, X., Gysel-Beer, M., Petäjä, T., Wang, J., and Schmale, J.: Automated identification of local contamination in remote atmospheric composition time series, Atmos. Meas. Tech., 15, 4195–4224, https://doi.org/10.5194/amt-15-4195-2022, 2022.

Beddows, D. C. S., Dall'Osto, M., and Harrison, R. M.: Cluster Analysis of Rural, Urban, and Curbside Atmospheric Particle Size Data, 43, 4694–4700, https://doi.org/10.1021/es803121t, 2009.

Bradley, R. S., Keimig, F. T., and Diaz, H. F.: Climatology of surface-based inversions in the North American Arctic, J. Geophys. Res., 97, 15699, https://doi.org/10.1029/92JD01451, 1992.

Francis, J. A., Vavrus, S. J., and Cohen, J.: Amplified Arctic warming and mid-latitude weather: new perspectives on emerging connections, Wiley Interdiscip. Rev. Clim. Chang., 8, e474, https://doi.org/10.1002/wcc.474, 2017.

Heslin-Rees, D., Burgos, M., Hansson, H.-C., Krejci, R., Ström, J., Tunved, P., and Zieger, P.: From a polar to a marine environment: has the changing Arctic led to a shift in aerosol light scattering properties?, 20, 13671–13686, https://doi.org/10.5194/acp-20-13671-2020, 2020.

Klonecki, A., Hess, P., Emmons, L., Smith, L., Orlando, J., and Blake, D.: Seasonal changes in the transport of pollutants into the Arctic troposphere-model study, 108, https://doi.org/https://doi.org/10.1029/2002JD002199, 2003.

Kolesar, K. R., Cellini, J., Peterson, P. K., Jefferson, A., Tuch, T., Birmili, W., Wiedensohler, A., and Pratt, K. A.: Effect of Prudhoe Bay emissions on atmospheric aerosol growth events observed in Utqiaġvik (Barrow), Alaska, 152, 146–155, https://doi.org/10.1016/j.atmosenv.2016.12.019, 2017.

Lee, M.-Y., Hong, C.-C., and Hsu, H.-H.: Compounding effects of warm sea surface temperature and reduced sea ice on the extreme circulation over the extratropical North Pacific and North America during the 2013–2014 boreal winter, Geophys. Res. Lett., 42, 1612–1618, https://doi.org/10.1002/2014GL062956, 2015.

Pernov, J. B., Beddows, D., Thomas, D. C., Dall´Osto, M., Harrison, R. M., Schmale, J., Skov, H., and Massling, A.: Increased aerosol concentrations in the High Arctic attributable to changing atmospheric transport patterns, npj Climate and Atmospheric Science, 5, 62, https://doi.org/10.1038/s41612-022-00286-y, 2022.

Quinn, P. K., Miller, T. L., Bates, T. S., Ogren, J. A., Andrews, E., and Shaw, G. E.: A 3-year record of simultaneously measured aerosol chemical and optical properties at Barrow, Alaska, J. Geophys. Res., 107, 4130, https://doi.org/10.1029/2001JD001248, 2002.

Schmale, J., Sharma, S., Decesari, S., Pernov, J., Massling, A., Hansson, H.-C., von Salzen, K., Skov, H., Andrews, E., Quinn, P. K., Upchurch, L. M., Eleftheriadis, K., Traversi, R., Gilardoni, S., Mazzola, M., Laing, J., and Hopke, P.: Pan-Arctic seasonal cycles and long-term trends of aerosol properties from 10 observatories, 22, 3067–3096, https://doi.org/10.5194/acp-22-3067-2022, 2022.

Schmeisser, L., Backman, J., Ogren, J. A., Andrews, E., Asmi, E., Starkweather, S., Uttal, T., Fiebig, M., Sharma, S., Eleftheriadis, K., Vratolis, S., Bergin, M., Tunved, P., and Jefferson, A.: Seasonality of aerosol optical properties in the Arctic, Atmos. Chem. Phys., 18, 11599–11622, https://doi.org/10.5194/acp-18-11599-2018, 2018.